# Smart Transportation Without Neurons - Fair Metro Network Expansion with Tabular Reinforcement Learning

## Abstract

We tackle the Metro Network Expansion Problem (MNEP), a subset of the Transport Network Design Problem (TNDP), which focuses on expanding metro systems to satisfy travel demand. Traditional methods rely on exact and heuristic approaches that require expert-defined constraints to reduce the search space. Recently, deep reinforcement learning (Deep RL) has emerged due to its effectiveness in complex sequential decision-making processes — it remains, however, computationally expensive, environmentally costly and require additional engineering to interpret. We show that MNEP problems are small enough to not require Deep RL methods. Reformulating the MNEP as a Non-Markovian Rewards Decision Process (NMRDP), we use tabular Q-Learning to achieve similar performance with significantly fewer training episodes, additionally offering greater interpretability. Additionally, we incorporate social equity criteria into the reward functions, focusing on efficiency and fairness, highlighting the versatility of our method. Evaluated in real-world settings — Xi'an and Amsterdam — our method reduces total episodes by a factor of 18 and total carbon emissions by a factor of 12 on average, while remaining competitive with Deep RL. This approach offers a replicable, modular, interpretable, and resource-efficient solution with potential applications to other combinatorial optimization problems.

## 1 Introduction

Public transport is fundamental to modern, fast-paced lifestyles, as it enables citizens to participate in employment, education, healthcare, and social activities (Martens, 2016). However, planning public transport networks is especially challenging due to physical, social, economic and legal constraints that complicate the creation of new transport routes, or the expansion of existing ones. Additionally, sustainability and equity are values that increasingly shape the design of public transport networks. Modern transport systems must be accessible, ensuring that citizens of all locations, socioeconomic statuses, and ages can benefit from these services (Martens, 2016). They also need to be efficient, as they must cover actual demand for mobility rather than being designed arbitrarily. Efficiency is also vital for sustainability: buses with low passenger loads can have a higher environmental impact per passenger than cars (Lowe et al., 2009), and low ridership can degrade the quality of transit systems over time (Mohring, 1972). These trade-offs add further complexity to transport design problems, leading to the need for increasingly sophisticated solutions.

The Transport Network Design Problem (TNDP) is an NP-hard combinatorial optimization problem that addresses the design of public transport, by maximizing total travel demand satisfaction (Farahani et al., 2013). For metro systems, a specific subset of TNDP, known as the Metro Network Expansion Problem (MNEP), is central to expanding existing metro lines within cities (Wei et al., 2020; Wang et al., 2023; Su et al., 2024). Metro networks are especially important in modern cities for their speed, reliability, and high passenger capacity compared to other traditional modes of public transport (Wang et al., 2023).

MNEPs focus on expanding an existing metro network in a city. Metro lines generally cover long distances, cross multiple urban zones, and are typically designed as relatively straight routes without excessive meandering (Wei et al., 2020). As a distinct sub-problem within TNDP, MNEP introduces additional constraints specific to metro network design.

Traditionally, TNDP problems have been approached with integer optimization and heuristic algorithms (Laporte & Pascoal, 2015; Owais & Osman, 2018), which require extensive expert-defined constraints to reduce the search space for tractability. Recently, the Metro Network Expansion Problem (MNEP) has been framed as a sequential decision-making problem, leveraging Reinforcement Learning (RL) to derive optimal solutions (Wei et al., 2020). RL is well-suited for sequential decision-making with multiple objectives, such as efficiency and fairness, and has been successfully applied to combinatorial optimization problems (Darwish et al., 2020; Raman et al., 2021; Jullien et al., 2022). Unlike traditional methods, RL can explore the search space flexibly by optimizing a reward function, avoiding the need for exponentially increasing constraints.

Given the large state-action spaces in many problems, the complexity of Reinforcement Learning (RL) may seem justified. Recently, Deep Reinforcement Learning (Deep RL) has shown promise in scaling combinatorial optimization, learning policy representations that autonomously identify key features and achieving state-of-the-art results in real-world problems (Mazyavkina et al., 2021; Neustroev et al., 2022; Xu et al., 2022).

While advances in computing power and algorithmic research suggest that RL could transform problems like MNEP, we argue that Deep RL is not always the ideal solution. Its substantial training time and environmental costs are becoming increasingly significant with the widespread deployment of AI systems (Anthony et al., 2020; Strubell et al., 2020; Patterson et al., 2021; Krishnan et al., 2022). Although MNEPs involve complex solution spaces, they are fundamentally static optimization problems with limited input features. Their scalability is inherently constrained—metro lines are typically spaced 1–3 kilometers apart (Giang et al., 2023) and are restricted in placement, shape, and other design factors. Complex neural network structures, which excel at capturing complex patterns in high-dimensional feature spaces, may not therefore be necessary for effective policy training. This is supported by findings in other machine learning domains (Cuccu et al., 2019).

In this paper, we argue that traditional RL methods can effectively tackle complex problems like MNEP when properly framed. We demonstrate that a tabular approach achieves competitive performance against deep-learning methods while significantly reducing training time in two real-world environments (Xi'an and Amsterdam). Additionally, our new formulation, in combination with tabular RL, offers greater interpretability than black-box deep-learning models.

To further showcase the potential of tabular RL, we explore social equity in MNEP by incorporating diverse reward functions based on various notions of social good. We extend the state-of-the-art RL formulation of MNEP to integrate fairness criteria. Our key contributions are the following: we reformulate the Transport Network Design and Metro Network Expansion problems as Non-Markovian Reward Decision Processes, significantly reducing the state-action space. We bridge machine learning and transport planning research by extending the RL framework to integrate considerations of social good, with both efficiency and fairness-based objectives. We propose a Monte Carlo Tabular Reinforcement Learning algorithm for MNEP, designed to require fewer training episodes than deep learning models. We validate our method in two real-world settings—Xi'an, China, and Amsterdam, Netherlands—demonstrating comparable performance to state-of-the-art Deep RL methods, with an 18-fold reduction in training episodes and a 12-fold reduction in $CO_2$ emissions. We provide all code, datasets, and hyperparameter settings to replicate our results and enable application to other combinatorial optimization problems[1].

The remainder of the paper is structured as follows: First, we position our work in the context of previous research (Section 2) and re-formulate the MNEP (Section 3). We continue by describing the tabular model and the proposed social-welfare reward functions (Section 4) and the real-world environments used in our experiments (Section 5). Finally, we present and discuss our results (Section 6).

## 2 Related Work

We outline previous work on the TNDP, reinforcement learning for combinatorial optimization, and the analysis of fairness in transportation.

---

[1]Github: https://github.com/*****/ (retracted for anonymous submission)

## 2.1 Transport Network Design Problem

Traditionally, the Transport Network Design Problem (TNDP) has been approached through a combination of integer optimization techniques and heuristic methods, including the use of pre-defined or dynamically discovered corridors (Laporte & Pascoal, 2015; Zarrinmehr et al., 2016; Gutiérrez-Jarpa et al., 2018), simulated annealing (Fan & Machemehl, 2006; Ahern et al., 2022), bee colony optimization (Yang et al., 2007; Szeto & Jiang, 2014), and genetic algorithms (Owais & Osman, 2018; Nayeem et al., 2018).

While these approaches have produced promising results in early studies, they have notable limitations. To make the problem tractable, they restrict the search space by either enforcing a long list of environment-specific constraints or by setting a predefined set of corridors. This restriction provides obstacles in application in large, real-world urban environments with diverse characteristics. More critically, narrowing the search space in this manner can exclude high-quality solutions that lie outside of these constraints.

## 2.2 Reinforcement Learning for Transport Network Design

Reinforcement Learning (RL) has proven effective for optimal long-term sequential decisions. Through straightforward reward mechanisms, an agent learns to understand its impact on the environment via trial-and-error, making RL well-suited for tackling real-world NP-hard combinatorial optimization tasks by leveraging demonstration and experience, without the need for expert prior knowledge (Mazyavkina et al., 2021; Wang & Tang, 2021; Bengio et al., 2021; Jullien et al., 2022; Darvariu et al., 2024). Although combinatorial optimization problems can also be approached with Supervised Learning (SL), recent studies have shown that RL can generalize more effectively than SL in common problems such as the Travelling Salesman Problem (Bello et al., 2017; Deudon et al., 2018) and Vehicle Routing (Nazari et al., 2018; Kool et al., 2018).

Despite the growing utility of RL in combinatorial optimization, its application to transport network design has only recently gained attention. Darwish et al. (2020) employed a policy gradient method to design bus lines, exploring the Pareto front between customer satisfaction and operational costs. Similarly, Wei et al. (2020) used a pointer-based model to address the Transit Network Design Problem (TNDP), demonstrating superior performance in demand satisfaction. More recently, Alkilane & Lee (2024) integrated Graph Neural Networks with a Monte Carlo Tree Search (MCTS) algorithm, leveraging network connectivity to enhance feature learning. Darvariu et al. (2023) also applied MCTS for graph expansion in existing metro networks, albeit without directly addressing the MNEP. Furthermore, Multi-objective Reinforcement Learning has been used in TNDP to balance efficiency with accessibility (Zhang et al., 2024; Michailidis et al., 2023).

Most work on the Transit Network Design Problem (TNDP) and the closely related Metro Network Expansion Problem (MNEP) has focused on complex deep reinforcement learning (Deep RL) models. This paper, however, challenges the necessity of such black-box models for problems where interpretability is crucial for decision-makers. We reformulate the problem to significantly reduce the action space without restricting the solution space, enabling a simpler, Monte Carlo-based tabular reinforcement learning approach. Our method is then benchmarked against the state-of-the-art Deep RL approach for MNEP (Wei et al., 2020).

## 2.3 Social Equity in Transport Network Design

Adopting notions of social equity in transport network design is challenging to optimize due to its multi-dimensional nature (Behbahani et al., 2019) and the inherent moral judgments involved (van Wee, 2011). Drawing on prior research in urban transport, we identify three key decisions necessary to incorporate fairness: utility measure, dimension, and fairness theory.

**Utility measure:** This is commonly achieved by establishing accessibility metrics, such as the number of reachable opportunities (Pereira et al., 2019; van der Veen et al., 2020; Hernandez, 2018), the affordability of accessing them (Farber et al., 2014), or a combination of both (El-Geneidy et al., 2016).

**Dimension:** Fairness can be assessed along spatial dimensions, where disparities are evaluated across different geographic or administrative units (Pereira et al., 2019; Delmelle & Casas, 2012), or through group-based measures, where groups are defined by socio-economic characteristics (e.g., income, race) (van der Veen et al., 2020; Pyrialakou et al., 2016; Cheng et al., 2021).

**Fairness theory:** Multiple theories of fairness and equity inform transport network design (Behbahani et al., 2019). Most approaches fall under horizontal fairness—aiming for equal utility across all units or groups—or vertical fairness, which prioritizes groups or areas in greater need (van Wee, 2011).

Despite these theoretical analyses, comprehensive application of fairness frameworks within machine learning for TNDP remains limited. Nonetheless, prior work has made initial attempts to integrate equity considerations. For example, Ramachandran et al. (2021) explore the efficiency-equity trade-off in graph augmentation using RL, applying their approach to Chicago's transport network (Ramachandran et al., 2021). Tedjopurnomo et al. (2022) compare bus line designs for advantaged and disadvantaged groups, though not using RL (Tedjopurnomo et al., 2022). Wei et al. (2020) account for equity by designing a weighted reward that balances travel demand with an area's development index, though this measure is implemented within the reward function and analyzed only minimally for its impact. The same approach is used by Zhang et al. (2024), who add one more component to the reward function.

Our paper presents the first attempt to bridge the gap between transport fairness research and RL-based transport network design in a comprehensive framework. We design fairness-based rewards based on Behbahani et al. (2019) definition, which targets an equitable distribution of benefits introduced by new transport lines. This framework is adaptable to various utility measures; in this study, we focus on Origin-Destination flows due to their relevance for mobility demand, rather than accessibility. Our analysis is done on a socio-economic group dimension, and we provide diverse reward functions that cover different fairness notions.

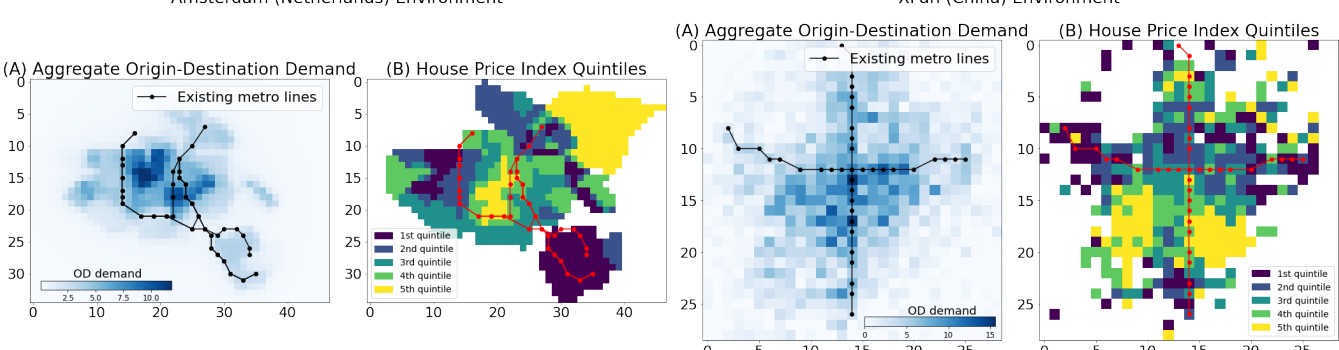

Figure 1: Two real-world case studies where the Metro Network Expansion Problem (MNEP) can be applied. The left side features Amsterdam, Netherlands, with each grid cell representing aggregate origin-destination demand (visualized using a blue colormap in panel A), along with the city's existing metro lines and housing price quintiles (panel B). On the right, similar data is displayed for Xi'an, China.

## 3 The Metro Network Expansion Problem

The Metro Network Expansion Problem (MNEP) is a subproblem of the Transport Network Design Problem (TNDP). Within the TNDP framework, the main objective is to expand the transport network by constructing a new line that maximizes the captured travel demand left unmet by the existing network.

In traditional formulations of TNDP and MNEP, the city is modeled as a two-dimensional grid environment with $n$ rows and $m$ columns, $H^{n \times m}$. The aim is to identify a set of adjacent cells $Z = \{z_1, z_2, \ldots, z_T \mid z_i \in H, \forall i = 1, 2, \ldots, T\}$, which sequentially connect to form a new metro line, in order to maximize the total captured demand. This demand is represented by an Origin-Destination (OD) matrix, $OD^{|H| \times |H|}$ (Guihaire & Hao, 2008; Farahani et al., 2013). Here, $OD[i, j]$ denotes the travel demand from grid cell $i$ to grid cell $j$. In the MNEP, the OD matrix is assumed to be symmetric and deterministic, remaining constant throughout the optimization process.

The size of set $Z$ is limited by a construction budget $B$, and a maximum number of stations $T$. We define a function $U(Z)$ that calculates the total added benefit of the generated line $Z$. In the traditional MNEP, $U(Z)$ is defined as the total sum of satisfied demand. The optimization problem is then defined as follows.

Find the set of connected cells $Z$, such that:

$$\max \quad U(Z) = \sum_i \sum_j OD[z_i, z_j], i \neq j$$
$$\text{s.t.} \quad cost(Z) \leq B \quad\quad\quad\quad\quad\quad (1)$$
$$|Z| \leq T$$

Here, the constraints $B$ and $T$ are strict, meaning that the new metro line must not exceed the specified budget or the total number of allowable stations.

The structural configuration of the metro line depends on the type of transport, which can be directed, as in bus or tram networks, or undirected, as is typical in metro systems. The focus of our paper is the design of metro networks, hence we tackle the Metro Network Expansion Problem (MNEP) (Wei et al., 2020).

### 3.1 Social Equity in the Metro Network Expansion Problem

The traditional MNEP primarily seeks to maximize total demand coverage, often overlooking the equitable distribution of benefits across various communities within the city. Prior work on reinforcement learning (RL) in this context also tends to prioritize efficiency and adopt a predominantly *utilitarian* approach (Wei et al., 2020). Here, we demonstrate that RL can effectively optimize for a wider array of objectives that encompass essential principles of social equity, as defined in transport planning literature. In addition to *utilitarianism* (Equation (1)), we emphasize two additional equity principles: *equal sharing of benefits* and *Rawlsian justice* as articulated by Rawls' theory of justice (Behbahani et al., 2019). Our focus centers on ensuring fairness in the allocation of satisfied Origin-Destination demand facilitated by the new line, paying particular attention to its distribution across different socioeconomic groups.

We first define a set of groups $G$, based on socioeconomic indicators such as income, development index, and education. Each cell $h \in H^{n \times m}$ in the environment is associated with a group $g \in G$. We adjust the objective function for each fairness notion accordingly, defining a utility function $U(Z, g)$ for each group $g \in G$, which returns the satisfied OD demand of line $Z$ for group $g$.

**Equal Sharing:** This egalitarian objective aims to equalize the added benefits of the transport line among groups in a city, commonly referred to as horizontal equity. In theory, equal sharing is achieved by minimizing the absolute differences between group utilities:

$$\min \sum_i \sum_j |U(Z, g_i) - U(Z, g_j)|, g_i, g_j \in G, i \neq j \quad\quad\quad\quad (2)$$

To implement fairness objectives in practice, we need to also incorporate total reward as, theoretically, Equation (2) could be minimized when all group utilities are 0. To address this, we encapsulate the equal-sharing notion using the Generalized Gini Index (GGI) (Siddique et al., 2020).

$$U(Z) = GGI(Z, W) = \sum_i^{|G|} W_i U(Z, \sigma(G)i), \qu\quad\quad\quad\quad (3)$$

where $\sigma$ is a permutation that sorts the groups in $G$ in descending order based on their utility prior to line creation, and $W_i$ are strictly decreasing weights (i.e., $W_1 > W_2 > \cdots > W|G|$) normalized to sum to 1.

**Rawls' Theory of Justice:** This approach aims to maximize benefits for the most disadvantaged group.

$$\max(U(Z, g_{min})), \qu\quad\quad\quad\quad\quad\quad\quad (4)$$

where $g_{min}$ represents the most disadvantaged group within $G$. In this paper, we define groups based on a house-price index as a proxy for area development, with $g_{min}$ as the group with the lowest house price index. Lower house price indexes are used as a proxy to identify the poorer areas of a city.

To apply this notion, we set the reward function as $U(Z) = U(Z, g_{min})$. In Figure 1, we illustrate the real-world cities of Amsterdam and Xi'an where we apply our method. We detail the environments in Section 5.

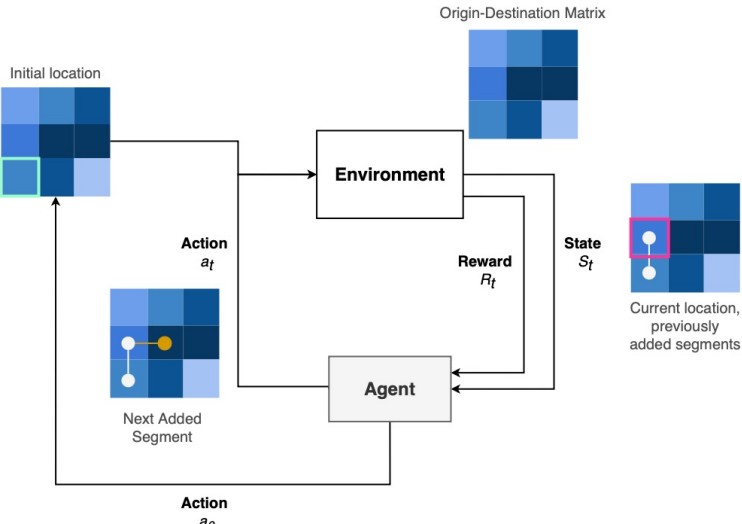

Figure 2: In the Metro Network Expansion Problem (MNEP), a reinforcement learning (RL) agent sequentially adds transport segments to the network. Each action represents the addition of a segment at a specific location, with rewards based on the demand met by that segment. The objective is to maximize the cumulative reward from all added segments.

## 4 Methods

We define the Metro Network Expansion Problem (MNEP) as a Non-Markovian Reward Decision Process (NMRDP) (Section 4.1) and describe the Tabular Q-Learning algorithm we use to solve it (Section 4.2).

### 4.1 Metro Network Expansion Non-Markovian Reward Decision Process

Recent approaches to the MNEP apply reinforcement learning (RL) by encoding each city grid cell as a potential action for the agent, resulting in an action space that scales linearly with the grid size ($|A| = |H|$) (Wei et al., 2020; Su et al., 2024), with a time complexity of $O(n \times m)$. While physical constraints mask certain actions to limit selectable cells at each timestep, this masking occurs only after the forward pass, immediately before the softmax layer (Wei et al., 2020; Su et al., 2024). As a result, the policy network must still process all potential cells in every state.

We argue that this complexity is unnecessary. Instead, we propose a two-stage approach: first, the agent selects a *starting cell*—the initial location for placing the first station on a metro line. The agent then navigates the grid by choosing among eight possible movement directions (north, south, east, west, and the four diagonal directions). Each movement forms a segment of the metro line, with the newly entered cell designated as the next station location.

With our approach, the initial cell selection and the subsequent episode steps are decoupled, substantially reducing the action space to 8, regardless of the grid size, reducing the time complexity at each step (except for the first) to $O(1)$. Additionally, we simplify the state representation to be the agent's current location, which can be efficiently encoded in a table with rows corresponding to the number of cells in the grid. However, this new formulation violates the Markov property, since the agent's current location alone does not encapsulate the previously placed stations. Future rewards depend on the sequence of past actions (Gaon & Brafman, 2020). Consequently, the decision process deviates from the Markov assumption that all necessary information is contained in the present state. Nonetheless, as in prior work addressing combinatorial optimization problems like the Travelling Salesman Problem (Bello et al., 2017; Kool et al., 2018), this departure from a strict Markovian framework is intentional and acceptable for our purposes. Our goal is to efficiently tackle the static MNEP by generating high-quality solutions rather than to satisfy all theoretical

properties of sequential decision making. And as we show in this paper, this relaxation does not lead to lower performance.

The Metro Network Expansion Problem (MNEP) can be formulated as a Non-Markovian Reward Decision Process (NMRDP), an extension of the Markov Decision Process (Gaon & Brafman, 2020), $\mathcal{M} = \langle \mathcal{S}, \mathcal{A}, \mathcal{P}, \mathcal{R}, \gamma, \mu \rangle$ as follows:

$\mathcal{S}$ is the state space, where each state $s_t = (x_t, y_t) \in \mathcal{S}$ represents the agent's current location in the two-dimensional city grid.

$\mathcal{A} = \{N, S, E, W, NE, NW, SE, SW\}$ is the action space, corresponding to the eight movement directions: North, South, East, West, and the four diagonals. The action taken at time $t$ is denoted as $a_t \in \mathcal{A}$.

$\mathcal{R} : \mathcal{S} \times \mathcal{A} \times \mathcal{S} \times \mathcal{H} \rightarrow \mathbb{R}$ is the reward function, which encodes the demand satisfied by constructing a metro line segment from $s_t$ to $s_{t+1}$. Since the reward depends on the history of visited states $\mathcal{H} = \{s_0, s_1, ..., s_t\}$, it is non-Markovian and cannot be fully determined by the current state-action pair alone. The reward received at time $t$ is $r_t = \mathcal{R}(s_t, a_t, s_{t+1}, \mathcal{H})$.

$\mu : \mathcal{S} \rightarrow [0, 1]$ is the probability distribution over the starting state $s_0$, which can be predefined, learned, or randomly sampled.

Given the discrete and episodic nature of the problem, we set the discount factor $\gamma = 1$, and the transition function $\mathcal{P}$ is deterministic. Figure 2 illustrates this formulation.

The action space in any state is further constrained by feasibility rules $F(Z_t)$, which enforce: no re-visiting of previously occupied cells, no movement beyond grid boundaries and no reversing direction or forming cycles. These constraints refine the set of allowable actions to adhere to the constraints of a metro line. More details on feasibility rules are provided in Appendix A, the accompanying code, and prior work (Wei et al., 2020; Zhang et al., 2024).

The reward function $\mathcal{R} : \mathcal{S} \times \mathcal{A} \times \mathcal{S} \times \mathcal{H} \rightarrow \mathbb{R}$ expresses the demand covered by the new metro segment, calculated in two steps. First, the direct demand between the new station and all previously existing stations on the line is computed (we use $Z_t$ to express the history $\mathcal{H}$ — the previously placed stations). Additionally, if connections between the new metro line and existing lines are identified, the reward is increased by the additional transfer demand between each station of the existing line and each station of the extended line (Wei et al., 2020). The total reward is the sum of these two components.

$$R_t = \underbrace{U(Z_t)}_{\text{direct demand}} + \underbrace{\sum_{l \in L} \mathbb{1}_{connect}(Z_t, l) \cdot U(l \times Z_t)}_{\text{transfer demand}}, \tag{5}$$

where $Z_t = z_1, ..., z_t$ is the set of all stations in the current line up to time $t$, $L$ is the set of all existing metro lines, $S_l$ is the set of stations in existing line $l$, $\mathbb{1}_{connect}(z_t, l)$ is an indicator function that equals 1 if station $z_t$ connects with line $l$ (shares a cell), and 0 otherwise.

## 4.2 Tabular Q-Learning for MNEP

We propose a tabular Q-learning algorithm for metro network expansion, in which a single reinforcement learning (RL) agent operates in two stages. We apply a Monte Carlo-based method to iteratively update the V and Q-tables through repeated environment interactions.

**Selecting the Initial Cell** An episode begins with the agent selecting the initial state $S_0$ (starting point for the metro line) using an $\epsilon$-greedy approach. When exploring, it picks a random cell; when exploiting, it picks the cell maximizing the expected return. The value of each cell as a starting position is given by $V_{\text{start}}(S_0) \in \mathbb{R}^{|H|}$, which estimates the expected return for beginning an episode at $S_0$.

**Action Selection and Transition** The agent selects actions using $\epsilon$-greedy approach. After choosing an action $A_t$, the agent observes a reward $R_t$ and deterministically transitions to a new state $S'$. This transition $(S_t, A_t, R_t)$ is stored in an episodic list, tracking the agent's path, which is later used to perform Monte Carlo

updates. Episodes end when one of three terminal conditions is met: (a) no available directions remain, (b) the budget is exhausted, or (c) the maximum number of allowed stations is reached.

**Monte-Carlo Returns and Policy Update** At the end of each episode, the agent updates the $V$ and $Q$-values using Monte Carlo estimation. First, the total discounted return, denoted by $J$ (we use $J$ here to avoid confusion with the group set $G$, departing slightly from standard RL notation), is calculated. Using this return $J$, the agent then updates the value functions accordingly.

$$Q(S_t, A_t) \leftarrow Q(S_t, A_t) + \alpha[J - Q(S_t, A_t)]$$
$$V_{\text{start}}(S_0) \leftarrow V_{\text{start}}(S_0) + \alpha[J - V_{\text{start}}(S_0)] \tag{6}$$

In Algorithm 1 we show the pseudocode of the proposed method.

---

**Algorithm 1** Tabular Metro Network Expansion with Monte-Carlo Updates

---

1: Parameters: $B$, $T$, $\alpha$, $\gamma$         ▷ Budget, total stations, RL parameters
2: Initialize $Q(s, a)$, $V_{\text{start}}$ for all $s$, actions $a$, empty $Episode$, $TotalCost \leftarrow 0$, and $ActionMask$ of ones.
3: **for** each episode **do**
4:     Select $S_0$ via $\epsilon$-greedy from $V_{\text{start}}$; add $S_0$ to $Z$
5:     **for** each step $t$ **do**
6:         Choose $A_t$ with $\epsilon$-greedy, considering $ActionMask$
7:         Execute $A$, receive reward $R$, observe next state $S'$
8:         Append $(S, A, R)$ to $Episode$, add $z_t$ to $Z$, update $TotalCost$, $ActionMask$, $S \leftarrow S'$
9:         **if** $SUM(ActionMask) = 0$ OR $TotalCost \geq B$ OR $t \geq T$ **then break**
10:         **end if**
11:     **end for**
12:     Initialize $J \leftarrow 0$
13:     **for** each step $(S_t, A_t, R_t)$ in $Episode$ from last to first **do**
14:         $J \leftarrow \gamma J + R_t$
15:         **if** $(S_t, A_t)$ is first in $Episode$ **then**
16:             $Q(S_t, A_t) \leftarrow Q(S_t, A_t) + \alpha(J - Q(S_t, A_t))$
17:         **end if**
18:     **end for**
19:     Update $V_{start}(S_0) \leftarrow \alpha(J - V_{start}(S_0))$
20:     Reset $TotalCost$, $Episode$, $Z$, and $ActionMask$
21: **end for**

---

## 5 Experiments

We ran and evaluated the model in two real-world case study cities: Xi'an and Amsterdam. To facilitate introducing directional constraints and to provide higher granularity, both cities are split into grids of equally-sized cells, rather than relying on census tracts (this assumption can be relaxed).

**Xi'an environment preparation**

Wei et al. (2020) created and publicly released the Xi'an environment [2]. The city is organized into a $H^{29 \times 29}$ grid, comprising $1km^2$ cells. An origin-destination (OD) demand matrix was generated from GPS data collected over one month from 25 million mobile phones. Each cell is linked to an average house price index — we categorize them to five quintiles to create groups. We selected the average house price as a proxy for neighborhood development, as it is widely available across various cities and raises no privacy concerns. While our group definitions rely on this metric, they could also incorporate other attributes, such as those based on protected categories. The environment already includes two existing metro lines, and our experiments focus on expanding the network by designing a third line. This setting provides a wealth of mobility demand data, contrasting with the case study in Amsterdam discussed below.

---

[2] https://github.com/weiyu123112/City-Metro-Network-Expansion-with-RL

**Amsterdam environment preparation**

The Amsterdam environment is organized into a $H^{35 \times 47}$ grid of $0.5km^2$ cells. This cell size was chosen to maintain similar problem complexity in all cities, taking into account the smaller size of Amsterdam. Since GPS data are unavailable, we estimate the origin-destination (OD) demand using the recently published universal law of human mobility, which indicates that the total mobility flow between two areas $i$ and $j$ is determined by their distance and visitation frequency (Schläpfer et al., 2021). We provide details on the estimation on Appendix B. As in the Xi'an environment, each cell is associated with an average house price sourced from the publicly available statistical bureau of the Netherlands [3]. The groups are defined as five quintiles based on this price.

## 5.1 Evaluation

We evaluate our proposed TabularMNEP algorithm against the state-of-the-art Deep Reinforcement Learning (DeepRL) method for Transport Network Design (Wei et al., 2020), as well as a Genetic Algorithm (GA) (Owais & Osman, 2018) and a Greedy Search Algorithm (GS) (Yang et al., 2007).

The methods are tested on four distinct reward functions: a utilitarian reward, maximizing total captured travel demand (Max Efficiency); two equal-sharing rewards using the Generalized Gini Index with weights of $1/2^i$ (GGI(2)) and $1/4^i$ (GGI(4)); and a Rawlsian reward that maximizes demand from the lowest house price quintile. We conducted a Bayesian hyperparameter search across 100 runs, selecting the top five configurations, running each five times, and choosing the one with the best average performance. More details on Appendix C. DeepRL was trained over 3,500 epochs (128 episodes per epoch, totaling 448,000 episodes), while TabularRL required only 25,000 episodes—a reduction of 18-fold in total training episodes.

To estimate emissions (kg $CO_2$ equivalent), we consider GPU electricity consumption (kWh), total training hours, and the carbon emissions per kWh based on the 2024 monthly average for `COUNTRY`[4], using the formula: `CO2 = Watt * TrainingHours * CarbonFactor`(*Lacost et al.*, 2019).

Model training used two types of in-house GPUs, the RTX 6000 Ada Generation (300 Watt) and GTX 1080Ti (250 Watt), depending on availability. Although our tabular method does not require a GPU, we report emissions based on GPU usage since a GPU-equipped node was reserved for model runs.

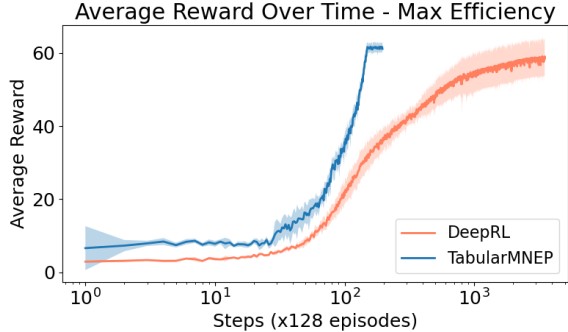

Figure 3: We demonstrate that the proposed TabularMNEP model achieves similar performance while requiring 18 times fewer episodes (x-axis is in log-scale).

# 6 Results

We ran both algorithms using 5 random seeds and provide code to replicate our results[5]. This section presents three key analyses: (1) a comparison of our proposed Tabular-TNDP method against recent approaches including Deep-RL, a Genetic Algorithm, and a Greedy Algorithm (Section 6.1); (2) a demonstration of

---

[3]https://www.cbs.nl/nl-nl/maatwerk/2019/31/kerncijfers-wijken-en-buurten-2019
[4]Country name retracted for anonymous submission.
[5]https://github.com/*****/****

| | Xi'an | | | | Amsterdam | | | |
|---|---|---|---|---|---|---|---|---|
| | Max. Efficiency | GGI(2) | GGI(4) | Rawls | Max. Efficiency | GGI(2) | GGI(4) | Rawls |
| Greedy Search (Laporte et al., 2005) | $33.8 \pm 0.00$ | $4.47 \pm 0.00$ | $2.16 \pm 0.00$ | $6.73 \pm 0.00$ | $6.17 \pm 0.00$ | $1.00 \pm 0.00$ | $0.38 \pm 0.00$ | $3.83 \pm 0.00$ |
| Genetic Algorithm (Owais & Osman, 2018) | $43.2 \pm 1.51$ | $6.05 \pm 0.22$ | $3.87 \pm 0.62$ | $11.07 \pm 0.85$ | $26.1 \pm 1.43$ | $2.66 \pm 0.16$ | $1.35 \pm 0.19$ | $8.80 \pm 0.68$ |
| DeepRL (Wei et al., 2020) | $62.7 \pm 2.86$ | $8.70 \pm 0.41$ | $5.63 \pm 0.88$ | $16.41 \pm 0.95$ | $35.7 \pm 0.06$ | $2.37 \pm 0.12$ | $0.76 \pm 0.01$ | $11.57 \pm 0.67$ |
| TabularMNEP (Ours) | $57.9 \pm 3.89$ | $8.13 \pm 0.58$ | $4.55 \pm 0.32$ | $14.65 \pm 1.05$ | $29.9 \pm 2.12$ | $2.53 \pm 0.39$ | $0.98 \pm 0.10$ | $9.75 \pm 1.10$ |

Table 1: Results on Xi'an and Amsterdam for 10 seeds.

| | Xi'an | | | | Amsterdam | | | |
|---|---|---|---|---|---|---|---|---|
| | Max. Efficiency | GGI(2) | GGI(4) | Rawls | Max. Efficiency | GGI(2) | GGI(4) | Rawls |
| DeepRL | 1.21 | 1.38 | 1.61 | 1.17 | 1.23 | 1.21 | 1.22 | 1.14 |
| TabularMNEP (Ours) | 0.05 | 0.13 | 0.13 | 0.12 | 0.06 | 0.28 | 0.28 | 0.10 |

Table 2: Estimated average emissions in kg $CO_2$ equivalent for each model's training.

TabularMNEP's versatility across multiple social-good rewards (Section 6.2); and (3) a justification for choosing TabularMNEP in scenarios where interpretability is crucial (Section 6.3).

## 6.1 TabularMNEP performs on par with DeepRL methods

Our proposed TabularMNEP method significantly outperforms both the Greedy Search (Laporte et al., 2005) and the Genetic Algorithm (Owais & Osman, 2018) baselines for most rewards. TabularMNEP achieves comparable performance to DeepRL across both the Xi'an and Amsterdam environments, considering both traditional and social good objectives defined in Section 3. Detailed averages and confidence intervals for all methods are presented in Table 1.

Notably, TabularMNEP achieves results within the confidence interval of DeepRL with substantially greater training efficiency, requiring only $25k$ episodes compared to DeepRL's $450k$ episodes (3500 epochs $\times$ 128 episodes). This $18\times$ reduction in training episodes is visualized in Figure 3 using a logarithmic x-axis.

In Table 2, we report the average $CO_2$ equivalent emissions from running our models across the four proposed reward functions. We observe that TabularMNEP requires, on average, $12\times$ fewer emissions to achieve performance comparable to the Deep RL baseline.

## 6.2 TabularMNEP effectively optimizes diverse rewards

As with Deep RL methods, TabularMNEP is capable of optimizing diverse rewards. Figure 4 shows the generated metro lines and the reward distribution among groups for both environments. The Max Efficiency reward function achieves the highest overall satisfied origin-destination flows, but we can observe that the rewards are distributed unequally among the five groups. In both Xi'an and Amsterdam, the highest quintiles exhibit greater satisfaction than the lowest quintiles, with inequality more pronounced in Amsterdam. This is due to the spatial distribution: in Xi'an, groups are more uniformly distributed, and segregation is lower, while in Amsterdam, the city center is dominated by higher-priced areas.

In contrast, the equality-based reward functions result in a more balanced distribution. Both GGI with $w = 2$ and $w = 4$ effectively equalize the rewards across groups. When $w = 4$, the rewards are distributed more equally, at the cost of overall efficiency. The Rawls reward prioritizes the lowest quintile in both environments, maximizing its satisfied demand. As intended, it directs the agent to optimize exclusively for the lowest quintile.

An additional insight from the Rawls reward function is its ability to reveal how isolated the lowest-utility group is. In Xi'an, maximizing for the lowest quintile creates "trickle-up" effects, benefiting other groups as well. However, in Amsterdam, where the lowest quintile is more segregated in the southeast, the generated line primarily benefits this group alone. This is further demonstrated in the spatial distribution of the lines, as shown in Figure 4.

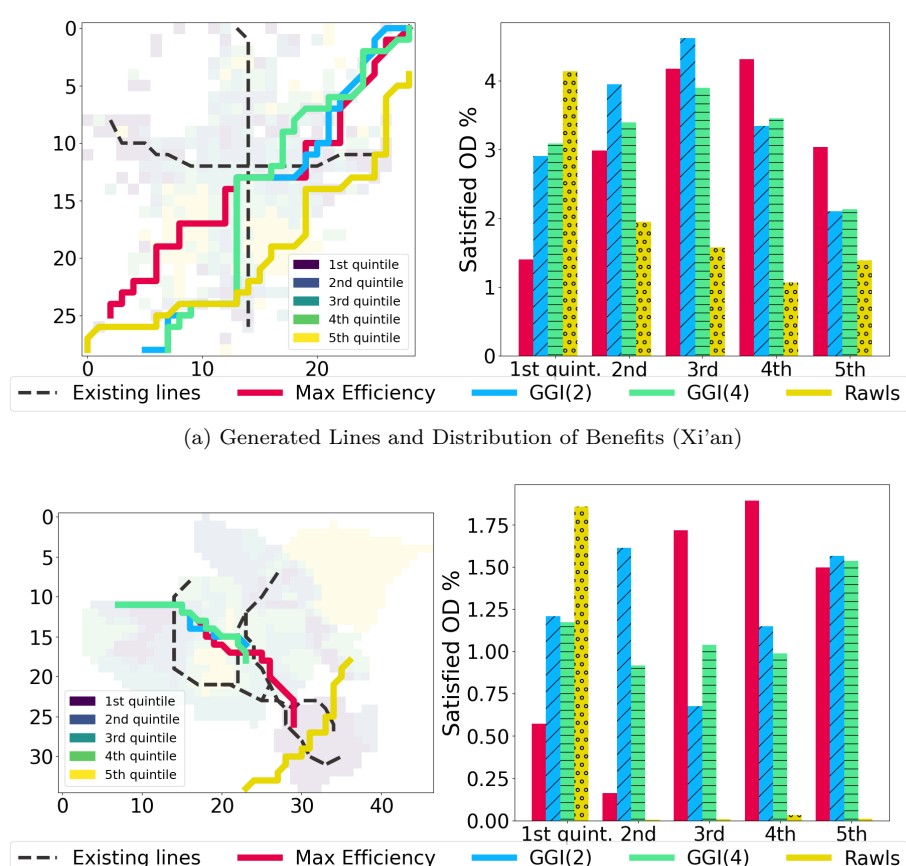

(a) Generated Lines and Distribution of Benefits (Xi'an)

(b) Generated Lines and Distribution of Benefits (Amsterdam)

Figure 4: We present the results of applying various reward functions to design transport lines in Xi'an (a) and Amsterdam (b). The left column displays the generated lines for each city, while the right column shows the distribution of satisfied demand across the five groups for the selected models.

### 6.3 Reduced state-space and TabularMNEP leads to more interpretable policies

Our new formulation, that reduces the state-space to be the grid, offers a key advantage in solving the Metro Network Expansion Problem (MNEP): inherent interpretability of the policies. As illustrated in Figure 5, we can visualize three critical aspects: (a) the optimal policy generating the metro line, (b) the average reward distribution across initial grid locations, and (c) the final Q-values with their corresponding best actions, which provide a direct interpretation for the best metro segment direction from each possible departing state. While similar visualizations could be produced for the previously proposed deep RL methods, there is a fundamental difference in how these values are stored and accessed. In Deep RL, policies are embedded within high-dimensional, latent representations, making it difficult to extract direct mappings from states to actions without additional processing, such as feature visualization or network probing. In contrast, our method explicitly stores values for each state-action pair, allowing for transparent inspection and direct modification, even during training. This interpretability provides decision-makers with insights beyond the model's output, enabling them to understand the relationship between actions and rewards, identify over-and under-explored areas in the city, and allowing genearting alternative routes to those produced by black-box models.

Transparency in this domain is particularly valuable as real-world metro planning often requires multiple alternative policies rather than a single solution. Additionally, tabular MNEP allows for incorporating spatial constraints after training, once the model has thoroughly explored the solution space. This post-

training constraint application enables the model's ability to discover diverse solutions, while still capable of accommodating practical limitations.

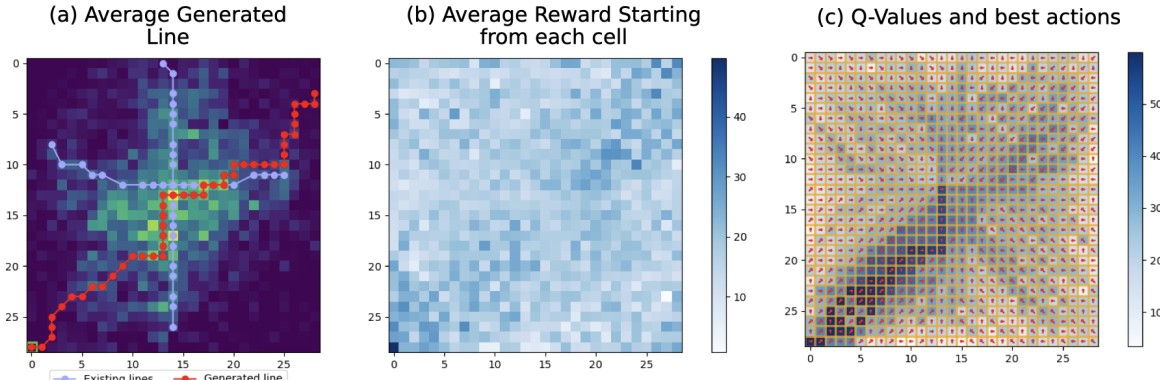

Figure 5: TabularRL provides better interpretability compared to DeepRL. In Panel (a), the metro line of a trained model optimized for maximum efficiency is illustrated. Panel (b) shows the average achievable reward from various starting points within the city, while Panel (c) displays the learned Q-values for each cell when the agent selects the action associated with the highest Q-value. Higher Q-values indicate more favorable locations for placing a metro station.

## 7    Conclusion

We demonstrate that simple, tabular-based reinforcement learning methods can effectively tackle complex combinatorial optimization problems with diverse objectives, such as the Transport Network Design and Metro Network Expansion problems. Our approach reformulates the problem to reduce the action space and employs distinct Q-tables for different action types.

We show that well-engineered problem reformulation, combined with established methods, can yield competitive results while requiring significantly less computational power. Our method runs efficiently on standard personal computers without a GPU and achieves performance comparable to state-of-the-art deep reinforcement learning techniques, despite using far fewer resources and requiring substantially less training time. Moreover, our approach enhances interpretability and flexibility in policy selection.

Our findings highlight that effective computational policy-making in real-world applications is achievable without relying on complex, black-box models. We hope this work encourages a re-evaluation of simpler models for other optimization challenges as well, such as link rewiring, which offers a similar setup (Yang et al., 2023). However, we acknowledge that tabular methods require encoding every possible state in the state space, which can pose scalability limitations. While our approach performs well in the Metro Network Expansion problem by constraining the state space, it may not generalize to problems with inherently large-scale state representations.

We would like to note that Reinforcement Learning in urban planning can enhance decision efficiency, but without careful consideration of the reward function, it can reinforce existing biases, favoring developed areas and deepening mobility inequities. Automated decision-making also risks reducing transparency and public engagement. Thus, the proposed models require human oversight, fairness considerations, and policy constraints for ethical deployment.

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

## A  Appendix – Feasibility Rules

The feasibility rules applied in this paper closely resemble those in previous studies (Wei et al., 2020; Zhang et al., 2024). The agent's actions are constrained using an `ActionMask`, which is updated at each timestep based on the agent's current location and prior positions. This approach ensures that the agent moves forward, avoids cyclical paths, and does not revisit locations where a station has already been placed.

Our method optimizes this process by maintaining a constant action mask length of 8, representing all possible directions (including diagonals), rather than the entire grid size. The agent's movement direction is established by its initial longitudinal and latitudinal steps. For example, if the agent begins by moving north, southward actions will be masked out to enforce forward progression. If the agent subsequently moves east, only actions corresponding to the north, east, and northeast directions remain available, with all other actions masked. Figure 6 illustrates how these feasibility rules are applied through the action mask during an episode.

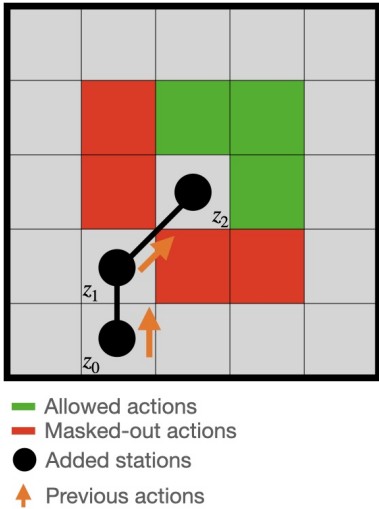

Figure 6: A snapshot of an episode, where the action mask created by feasibility rules constraints the next available actions to the agent.

## B  Appendix – Amsterdam environment preparation

GPS data is unavailable for Amsterdam, so we estimate the origin-destination (OD) demand using the recently published universal law of human mobility, which indicates that the total mobility flow between two areas $i$ and $j$ is determined by their distance and visitation frequency (Schläpfer et al., 2021). The calculation is as follows:

$$OD_{ij} = \mu_j \mathsf{K}_i / d_{ij}^2 \ln(f_{max}/f_{min}) \tag{7}$$

Here, $\mathsf{K}_i$ is the total area of the origin location $i$, $d_{ij}^2$ is the (Manhattan) distance between $i$ and $j$, and $\mu_j$ represents the magnitude of flows, computed as:

$$\mu_j \approx \rho_{pop}(j) rad_j^2 f_{max} \tag{8}$$

Where $rad_j^2$ is the radius of area j. We estimate the flows over a week by setting $f_{min}, f_{max}$ to 1/7 and 7 respectively. The grid cells are of equal size, $K_i$ and can be omitted from the calculation.

## C  Appendix – Hyperparameter Tuning and Selection

**Greedy Search (GS)** — Greedy search is a simple greedy algorithm that begins by adding the segment with the largest OD flow, and then greedily expanding the network from this segment, while following the feasibility rules.

**Genetic Algorithm (GA)** — We conducted experiments using two sets of hyperparameters: one based on Wei et al., which included a population size of 500, and crossover and mutation probabilities of 0.9, and another set based on Owais & Osman, which used a population size of 500, a crossover probability of 0.6, and a mutation probability of 0.05. We chose the second set of hyperparameters, as they are directly taken from the original source and are more commonly used in Genetic Algorithms. To ensure a fair comparison with the Tabular Q-learning method, we trained the Genetic Algorithm for a total of 25,000 episodes, which consisted of 50 iterations, each with 500 generated solutions.

**Deep Reinforcement Learning (DeepRL)** — We conducted experiments using the hyperparameters reported by Wei et al., which, at the time of writing the paper, were considered state-of-the-art methodology. The hyperparameters are as follows:

- Hidden size: 128

- Static size: 2

- Dynamic size: 1

- Number of layers: 1

- Dropout: 0.1

- Max epochs: 3500

- Training size: 128

- Actor learning rate: 0.001

- Critic learning rate: 0.001

**Tabular Q-learning (TabularMNEP)** — To select the hyperparameter values for TabularMNEP, we performed multiple sweeps of parameters and methods, using a bayesian optimization approach, via the Weights & Biases library [6]. In Table 3 we show the ranges we used for each hypeparameter. Additionally to the Monte-Carlo update method we used in our final experiments, we also tried Temporal-Difference and Upper Confidence Bound methods. In the code we provide the commands to replicate our results.

For **Xi'an**, the final experiments were conducted with 47 stations. The Max. Efficiency setting used a single group, while the Rawls, GGI4, and GGI2 settings used five groups (for calculating fairness). All experiments ran for 25000 training episodes with an epsilon decay over 16000 steps, a warmup of 3000 steps, and a single test episode. The initial and final epsilon values were set to 1 and 0.01, respectively, with a learning rate ($\alpha$) of 0.1 and a discount factor ($\gamma$) of 1. Exploration followed an epsilon-greedy strategy, and updates were done via Monte Carlo methods.

For **Amsterdam**, the final experiments were conducted with 21 stations. The Max. Efficiency experiments used a single group, with 14000 epsilon decay steps, 0 warmup steps, and learning rate ($\alpha$) set to 0.1. The Rawls, GGI4, and GGI2 experiments used five groups, with epsilon decay set to 14,000 steps and no warmup steps. All experiments ran for 25,000 training episodes, with an initial epsilon of 1, a final epsilon of 0.01, a discount factor ($\gamma$) of 1. The exploration followed an epsilon-greedy strategy, and updates were also performed using Monte Carlo methods.

---

[6]https://wandb.ai/site/

| Parameter | Values |
|---|---|
| alpha | 0.1, 0.2, 0.3, 0.4, 0.5, 0.6, 0.7, 0.8, 0.9, 1 |
| final_epsilon | 0.05, 0.1, 0.01 |
| gamma | 0.0, 0.7, 0.8, 0.9, 0.95, 0.99, 1 |
| initial_epsilon | 1, 0, 0.1, 0.2, 0.3, 0.4 |
| train_episodes | 1000, 2000, 4000, 5000, 7000, 10000, 15000, 20000, 25000, 30000 |
| epsilon_warmup_steps | 0, 200, 500, 1000, 2000, 3000, 4000, 5000, 6000, 7000, 9000, 10000 |
| epsilon_decay_steps | 2000, 3000, 4000, 5000, 6000, 7000, 8000, 10000, 12000, 14000, 16000, 18000, 20000 |
| exploration_type | egreedy, ucb, egreedy_constant |
| q_initial_value | 0, 20, 40, 60, 80, 100, 120, 140 |
| V_start_initial_value | 0, 20, 40, 60, 80, 100, 120, 140 |
| ucb_c_q | 0, 2, 4, 8, 16, 32, 64, 128 |
| ucb_c_qstart | 0, 2, 4, 8, 16, 32, 64, 128 |
| update_method | td, mc |

Table 3: Parameter values tried during hyperparameter search.

