# OpenReview forum: "Smart Transportation Without Neurons - Fair Metro Network Expansion with Tabular Reinforcement Learning"
_TMLR — Rejected by TMLR_

### Review · Reviewer_FpTH · 2024-12-14

**Summary Of Contributions:**

This paper considers the Metro Network Expansion Problem, i.e., the task of adding new lines to a metro transportation network so as to improve an objective function such as its ability to satisfy origin-destination demands. The work proposes using a tabular reinforcement learning approach, unlike recent papers on this topic that opt for Deep RL methods, basing this on the premise that the learning process can be made more efficient and interpretable. Furthermore, a number of objective functions are considered, which go beyond merely improving traffic satisfaction and treat the fairness of the network expansion decisions in terms of impacts on groups associated with each region. The authors conduct experiments on two metro networks, which validate that the method can perform comparably to a Deep RL technique while requring less computational resources. An analysis of the resulting policy is also carried out, which shows that the technique possesses interpretability.

**Audience:**

Yes

**Claims And Evidence:**

Yes

**Requested Changes:**

Please see below for a list of suggestions that I believe would improve the work. However, I am uncertain that revisions that do not involve a substantial redesign of the algorithm would secure an acceptance recommendation of the work, since algorithms exist that simply do better in achieving the stated goals of the paper. Currently, the key message of the paper seems to be "you don't need a neural network if the MDP is sufficiently small" and "tabular RL can be more efficient for small problems", which I do not view as substantial enough contributions to warrant acceptance. Another core suggestion would be to focus the paper and its narrative on the analysis of the different objective functions, since this is the main novelty of the work.

### Major suggestions
- Please address W2 and W3 above, first of all, to ensure the validity of the results and interpretation.
- Related work: papers [1] and [2] should be added to the discussion and the findings / decisions interpreted in the light of point W1 above. Another recent work on RL for metro network expansion that should be cited is [3]. [4] is an RL approach for another related network design problem.
- The MDP is not precisely defined in mathematical terms, and it definitely should be. There are also several inaccuracies, e.g., in 4.1 $\mathcal{S}$ is *not* the agent's current location (an individual state), but the state space, which will contain the initial network and all additions made up to this point.
- Sections 4.1 and 4.2 contain a mix of MDP element definitions, algorithm formulation, various implementation details, and hyperparameter settings. Consider separating these clearly.

### Minor suggestions
- $Q(S_0)$ implies that there is no action, but it actually indicates the value of picking each of the $H$ squares as the intial regions. It would be cleaner to explicitly define two Q-functions e.g. $Q_1$, $Q_2$.
- Citation style: distinguish between uses of `\citet` and `\citep`
- In Figure 3, y-axes should indicate the metric (rather than putting them in the plot title)

### References
[1] Darvariu, V. A., Hailes, S., & Musolesi, M. (2024). Graph Reinforcement Learning for Combinatorial Optimization: A Survey and Unifying Perspective. Transactions on Machine Learning Research (TMLR). https://openreview.net/forum?id=HduK51xNtS

[2] Darvariu, V. A., Hailes, S., & Musolesi, M. (2023). Planning spatial networks with Monte Carlo tree search. Proceedings of the Royal Society A, 479(2269), 20220383.

[3] Alkilane, K., & Lee, D. H. (2024). MetroZero: Deep Reinforcement Learning and Monte Carlo Tree Search for Optimized Metro Network Expansion. IEEE Transactions on Intelligent Transportation Systems.

[4] Yang, S., KAILI, M., Wang, B., Yu, T., & Zha, H. (2023). Learning to boost resilience of complex networks via neural edge rewiring. Transactions on Machine Learning Research (TMLR). https://openreview.net/pdf?id=moZvOx5cxe

**Strengths And Weaknesses:**

## Strengths

**S1**. In my view, the paper correctly identifies that Deep RL is unnecessary at the scale of the considered problems, and tabular techniques can perform more efficiently.

**S2**. In comparison to other RL-based works, its analysis of several objectives that take fairness properties into account is interesting and perhaps closer to criteria that policy-makers would adopt.

## Weaknesses
**W1. Simplistic algorithm**
- The proposed technique is simplistic and likely to scale poorly as the size of the problem grows. If the best computational efficiency for finding a solution to a specific problem instance is the goal, then model-free RL is unlikely to be the best option.
- Even in the realm of model-free RL, Q-learning style algorithms would show better sample complexity than the approach presented. But given that the MDP model is deterministic and can be formulated analytically in this case, model-based decision-time planning techniques would be a better choice over model-free RL, a point also made in a recent TMLR survey paper on RL for combinatorial optimization [1].
- In fact, the paper [2] treats a related metro network design problem (albeit with a different formulation). It shows that Monte Carlo Tree Search can obtain substantially better scalability to large metro systems than Deep RL while also not using a neural network. It also adopts a two-stage action space (first choose the origin of the connection, then the destination) similar to the one proposed here. In my view, MCTS-style algorithms would align better with the goals of the authors.

**W2. Weaknesses in the evaluation**
- Non-RL baselines are not implemented and results are reported directly for a single objective. All methods should be evaluated for all objectives, and undergo the same hyperparameter tuning regime.
- The use of 5 seeds for evaluation makes (nearly all) confidence intervals overlap, 10 is a bare minimum that should be adopted.

**W3. Claims on reward diversity and interpretability**
- Section 6.2 highlights that "TabularMNEP is capable of optimizing diverse rewards", but so can the Deep RL method, so this might be interpreted to be misleading.
- Section 6.3 claims that the method leads to more interpretable policies, but I see no reason why the exact same analysis cannot be applied with the Deep RL method? This relies on average rewards and Q-values, which can also be extracted from a neural net, and while they may be somewhat more accurate, it is misleading to claim that this method has more interpretability.

---

> ### Author Response · Authors · 2025-02-20
> **Response to review FpTH (Part 1)**
>
> We thank the reviewer for taking the time to read our paper and to provide a valuable and very helpful review. We feel encouraged to read that the reviewer acknowledges the value of considering fairness criteria and the practical relevance of our approach for policy-making.
> While the reviewer raises valid concerns about scalability and the deterministic nature of the MDP, we believe that our approach is not inherently flawed, but rather represents an alternative, model-free approach.
>
> Please find our point-by-point response below:
>
> **W1 --- Simplistic Algorithm**
>
> **Comment**: The proposed technique is simplistic and likely to scale poorly as the size of the problem grows. If the best computational efficiency for finding a solution to a specific problem instance is the goal, then model-free RL is unlikely to be the best option.
>
> **Response:** We respectfully disagree with the assessment that the simplicity of our Q-learning-based method is a weakness. On the contrary, we demonstrate that a model-free, tabular Q-learning approach performs on par with Deep RL while requiring significantly fewer episodes and consuming considerably less energy. This efficiency is a key strength of our method.
>
> On scalability, we test our Q-learning algorithm on grids of 841 and 1645 cells, with the latter corresponding to a 0.5 km station spacing---an unusually small distance for urban metro lines, which are typically spaced around 1--3 km apart. There is an inherent limitation in how much scale can be meaningfully introduced into the metro network design problem. Additionally, we impose metro network constraints that significantly reduce the action space, ensuring that the learning process remains computationally feasible even as network size increases. In fact, paper [2] referenced in your comment mentions "*Due to computational budget constraints, we limit the sizes of networks considered to $|V| = 150$*". So their method is tested on a much lower scale than ours.
>
> **Comment:** Even in the realm of model-free RL, Q-learning style algorithms would show better sample complexity than the approach presented. But given that the MDP model is deterministic and can be formulated analytically in this case, model-based decision-time planning techniques would be a better choice over model-free RL, a point also made in a recent TMLR survey paper on RL for combinatorial optimization [1].
>
> **Response:** While model-based methods can achieve lower sample complexity, we argue that sampling from a simulated metro network design environment is not inherently expensive. In contrast, model-based methods require additional computation to learn and update the model. Although such methods may reduce the number of episodes needed, this is not a meaningful advantage in our setting, as episodes themselves are not computationally a problem. Furthermore, model-based approaches such as MCTS are also notorious for scaling poorly [4].
>
> **Comment:** In fact, the paper [2] treats a related metro network design problem (albeit with a different formulation). It shows that Monte Carlo Tree Search can obtain substantially better scalability to large metro systems than Deep RL while also not using a neural network. It also adopts a two-stage action space (first choose the origin of the connection, then the destination) similar to the one proposed here. In my view, MCTS-style algorithms would align better with the goals of the authors.
>
> **Response:** Paper [2] mentions "*Due to computational budget constraints, we limit the sizes of networks considered to $|V | = 150$*". Their method is tested on a much lower scale than ours, which is tested up to a grid of $|V | =1645$ nodes.
>
> On generalization: our Q-learning-based approach also has the potential to generalize to stochastic optimization settings, such as variations in origin-destination flows during peak and off-peak hours. This is an area where model-based methods, like MCTS, would struggle. Moreover, our approach can be extended to incorporate a learned model for planning steps between learning iterations---such as in a Dyna-Q framework [5]---offering a hybrid model-based alternative while retaining the advantages of Q-learning and further reducing sample complexity. We will make sure to add these potential extension ideas in the final version of our paper, and we thank very much the reviewer for sparking these ideas.
>
> Our approach enables modularity, allowing for the modules to be independed from each other. E.g. the first stage, where the agent selects a starting location, is a separate process that can be done with a neural network to allow scalability. The sequential process afterward is more simple and does not suffer from scaling issues.
>
> In summary, we believe our method provides a simple but effective solution to the metro network design problem, balancing efficiency with strong performance. Given the specific problem that we tackle, we trust simplicity is an advantage rather than a limitation.

---

> > ### Author Response · Authors · 2025-02-20
> > **Response to review FpTH (Part 2)**
> >
> > **W2 --- Evaluation**
> >
> > **Comment**: Non-RL baselines are not implemented and results are reported directly for a single objective. All methods should be evaluated for all objectives, and undergo the same hyperparameter tuning regime.
> >
> > **Response**: We have implemented both the Genetic Algorithm baseline and a Greedy Search algorithm and ran experiments for all reward functions on both environments. We observe that our original claims still hold, and we have updated the paper accordingly.
> >
> > **Comment**: The use of 5 seeds for evaluation makes (nearly all) confidence intervals overlap, 10 is a bare minimum that should be adopted.
> >
> > **Response:** We have now ran every algorithm for 10 seeds and updated our results in the paper. All our conclusions remain.
> >
> > **W3 --- Diversity and Interpretability**
> > **Comment**: Section 6.2 highlights that "TabularMNEP is capable of optimizing diverse rewards", but so can the Deep RL method, so this might be interpreted to be misleading.
> >
> > **Response:**
> > Our claim does not suggest that Deep RL methods cannot optimize diverse rewards, but rather highlights that TabularMNEP is as good as Deep RL, while requiring less computation. We have changed the title of the section to "TabularMNEP effectively optimizes diverse rewards", and also added a clarifying statement in the beginning of the section "As with Deep RL methods, TabularMNEP is capable of optimizing diverse rewards".
> >
> > **Comment**: Section 6.3 claims that the method leads to more interpretable policies, but I see no reason why the exact same analysis cannot be applied with the Deep RL method? This relies on average rewards and Q-values, which can also be extracted from a neural net, and while they may be somewhat more accurate, it is misleading to claim that this method has more interpretability.
> >
> > **Response:** While Deep RL methods can, in theory, extract Q-values, their interpretability differs significantly from traditional approaches. Please note that the challenges of interpreting Deep RL -- in opposition to tabular RL -- have been previously identified [3]. Neural networks do not store explicit state-action value mappings in an accessible way; instead, they encode them in high-dimensional latent spaces, making it hard to understand how specific inputs influence decisions. Extracting this information requires post-hoc analysis for each state-action pair, typically through saliency maps or t-SNE plots [3], which are sensitive to simple transformations [1] and generally unreliable [2]. In our approach, instead, all that is required is imply to store the Q-table after training.
> >
> > Furthermore, in deep Q-networks, Q-values are often estimated using probability distributions over actions, requiring additional steps to devise a single, interpretable value. In contrast, TabularMNEP maintains an explicit tabular representation of the best estimation of the current Q-values, allowing for direct inspection and interpretation. The ability to directly query state-action values enables straightforward policy evaluation, even by non-experts, which is not as readily feasible with Deep RL.
> >
> > Furthermore, our method inherently allows one to identify underexplored areas of the state space by observing Q-value distributions, which is significantly harder with deep models. This offers flexibility in the designer/decision-maker for further post-training adjustments. We will add these clarifications, regarding the advantages of TabularMNEP on interpretability, on the final version of our paper.
> >
> > **Comment**: Another core suggestion would be to focus the paper and its narrative on the analysis of the different objective functions, since this is the main novelty of the work.
> >
> > **Response:** This is an excellent point! We will definitely highlight the additional core contribution of our paper, related to considering different objective functions.
> >
> > **Comment:** Related work: papers [1] and [2] should be added to the discussion and the findings / decisions interpreted in the light of point W1 above. Another recent work on RL for metro network expansion that should be cited is [3]. [4] is an RL approach for another related network design problem.
> >
> > **Response:** We have adapted the paper to discuss papers [1, 2, 3]. Thank you for providing these references. We believe [4] can inspire new directions on network rewiring; however, we struggled to connect the approach proposed with network expansion (the core aim of our paper). We refer it as a connection with potential future work. %is not relevant to the problem we tackle, as it concerns an edge rewiring problem, different to the MNEP.
> >
> > **Comment:** The MDP is not precisely defined in mathematical terms, and it definitely should be.
> >
> > **Response:** We have revamped the formulation in Section 4.1 to make it more precise and add more details.

---

> ### Author Response · Authors · 2025-02-20
> **Response to review FpTH (Part 3)**
>
> **Comment:** There are also several inaccuracies, e.g., in 4.1  is not the agent's current location (an individual state), but the state space, which will contain the initial network and all additions made up to this point.
> **Response:** We have revamped section 4.1 to clarify our formulation, which is an MDP with non-markovian rewards. We also further updated 4.1 to add more discussion on this choice.
>
> **Comment:** Sections 4.1 and 4.2 contain a mix of MDP element definitions, algorithm formulation, various implementation details, and hyperparameter settings. Consider separating these clearly.
>
> **Response:** We have moved the reward definition and calculation paragraph on 4.1, to align with the MDP formulation. We have also removed specific mentions of parameter values from 4.1
>
>
> **Comment:** Q(S0) implies that there is no action, but it actually indicates the value of picking each of the squares as the intial regions. It would be cleaner to explicitly define two Q-functions e.g. Q1 Q2.
>
> **Response:** We have revamped the value functions notation. We now denote table that holds the values of the initial state as $V_{start}$, which indicates the value of the state as an intial state. Since the fist action selects a state, we believe this is an appropraite notation for the initial cell. We didn't opt for the Q1, Q2 option to avoid confusion with double Q-learning methods, that adopt this notation.
>
> **Comment:** Citation style: distinguish between uses of citet and citep.
>
> **Response:** We have updated the citations to distinguish between citet and citep across the paper.
>
> **Comment:** In Figure 3, y-axes should indicate the metric (rather than putting them in the plot title)
>
> **Response:** We have removed the figure, as it conveyed the same information with Table 1.
>
> Thank you once again for the thoughtful review, which definitely allowed us to improve our paper!
>
> **References**
>
> [1] Kindermans, P. J., Hooker, S., Adebayo, J., Alber, M., Schütt, K. T., Dähne, S., ... \& Kim, B. (2019). The (un) reliability of saliency methods. Explainable AI: Interpreting, explaining and visualizing deep learning, 267-280.
>
> [2] Annasamy, R. M., \& Sycara, K. (2019, July). Towards better interpretability in deep q-networks. In Proceedings of the AAAI conference on artificial intelligence (Vol. 33, No. 01, pp. 4561-4569).
>
> [3] Zahavy, T., Ben-Zrihem, N., \& Mannor, S. (2016, June). Graying the black box: Understanding dqns. In ICML 2016 (pp. 1899-1908). PMLR.
>
> [4] Liu, A., Chen, J., Yu, M., Zhai, Y., Zhou, X., \& Liu, J. (2018). Watch the unobserved: A simple approach to parallelizing monte carlo tree search. arXiv preprint arXiv:1810.11755.
>
> [5] Richard S. Sutton. 1991. Dyna, an integrated architecture for learning, planning, and reacting. SIGART Bull. 2, 4 (Aug. 1991), 160–163. https://doi.org/10.1145/122344.122377

---

> > ### Comment · Reviewer_FpTH · 2025-03-05
> > **Response to authors**
> >
> > Thanks for your responses. A few points I'd like to highlight:
> >
> > - On scalability: decision-time planning approaches like MCTS are substantially more scalable than the approach presented, which effectively solves the MDP exhaustively. MCTS (applied online) only solves the MDP from the current state of interest, and is able to sample to estimate returns instead of expanding every state. This comparison relies on the MDP being the same. It is indeed the case that if you blindly compare the number of nodes in the work mentioned above that applies MCTS _and_ this paper, which has a significantly reduced state and action space, the comparison will not be equitable. If you were to apply MCTS to your MDP definition, you would certainly reduce the amount of computation required.
> > - In fact, you do not need to learn a model at all to apply MCTS, since it can be formulated analytically.
> > - The paper has improved substantially in presentation (e.g., addition of MDP definition) and soundness (e.g., addition of baselines, multiple seeds).
> > - I am still skeptical that the paper presents a sufficient contribution in either methodological terms or the considered application to warrant publishing in this venue.

---

> > > ### Author Response · Authors · 2025-03-07
> > > **Response to additional points by Reviewer FpTH**
> > >
> > > We sincerely appreciate your thoughtful feedback and constructive criticism, and your recognition of the improvements of the technical soundness in our latest version.
> > >
> > > On scalability: Our work introduces a new, interesting problem formulation for the MNEP—which opens avenues for more research on the field— and applies a tabular method that already demonstrates significant gains in computational complexity compared to previous approaches. We acknowledge that MCTS could be used to for even more efficiency, and future research could explore it.
> > >
> > > On sufficient contribution: We understand your skepticism. Nonetheless, we believe our work makes a valuable and interesting contribution by introducing a) a new formulation, b) comprehensive and sound evaluation in two real-world cities, and c) integration of fairness-based reward functions. This work opens avenues for further ML research, including additional advancements on the MNEP. We will adapt the final version of the paper to mention MCTS as a logical next step in this framework.
> > >
> > > We greatly appreciate this constructive discussion and your insightful feedback, which have helped refine and improve our paper. Thank you again for your time and engagement.

---

### Review · Reviewer_NMVQ · 2025-01-21

**Summary Of Contributions:**

The paper demonstrates that a simple tabular reinforcement learning algorithm can effectively address the complex combinatorial optimization problem of Transport Network Design and Metro Network Expansion. The findings are based on case studies of transport networks in two cities: Amsterdam (Netherlands) and Xi'an (China). Experimental results show that the tabular approach achieves performance with at least an order of magnitude higher efficiency than Deep Learning RL methods.

**Audience:**

Yes

**Broader Impact Concerns:**

no concerns

**Claims And Evidence:**

Yes

**Requested Changes:**

I may have missed something, but I would appreciate an explanation of why the paper is written as if presenting tabular Q-learning for TNDP and MNEP, instead of explaining that often TNDP and MNEP problems are small enough to allow for tabular Q-learning to be efficient but as or more accurate than more computationally costly DeepRL.

**Strengths And Weaknesses:**

## strength

The paper effectively demonstrates a practical application by solving the Transport Network Design Problem (TNDP) and Metro Network Expansion Problem (MNEP) using a tabular formulation of Q-learning. This approach is notably more efficient than utilizing Deep Reinforcement Learning (DeepRL) methods.

## weakness

- The paper does not adequately explain why the superior efficiency of the tabular Q-learning approach over DeepRL is surprising, given that tabular Q-learning provides an exact solution, while DeepRL offers an approximation.
- Furthermore, the paper does not address the inherent limitations of the tabular Q-learning approach, such as its lack of scalability and high memory requirements.

---

> ### Author Response · Authors · 2025-02-20
> **Response to Reviewer NMVQ**
>
> We thank the reviewer for taking the time to provide a constructive and helpful review, and we appreciate the recognition of our method's efficiency. Please find our point-by-point review below:
>
> **Comment:** The paper does not adequately explain why the superior efficiency of the tabular Q-learning approach over DeepRL is surprising, given that tabular Q-learning provides an exact solution, while DeepRL offers an approximation.
>
> **Response:** We would like to clarify that tabular Q-learning is also an approximation, as it is run for a specific, finite number of timesteps. Our paper aims to demonstrate that DeepRL is not necessary for tasks like the Transport Network Design Problem (TNDP) and Metro Network Expansion Problem (MNEP), as tabular alternatives can achieve comparable performance with significantly less computational cost. We provide evidence of this by showcasing the performance of our approach on large-scale, high-granularity environments such as the cities of Xi’an and Amsterdam. As we discussed in our response to Reviewer FpTH, we push the limits of scalability and granularity in the context of the Metro Network Design Problem, and our TabularMNEP approach performs well in such scenarios.
>
> **Comment:** Furthermore, the paper does not address the inherent limitations of the tabular Q-learning approach, such as its lack of scalability and high memory requirements.
>
> **Response:** We have now added a statement in the conclusion that clarifies that tabular Q-learning is effective in problems where the state space can be reduced to sensible levels to be represented in a table, but that it may not generalize to problems with inherently high-dimensional state spaces. We shall also highlight that we apply our method to a large-scale problem, with a number of nodes (i.e., stations) and edges characterizing a real-world scenario.
>
> **Comment:** I may have missed something, but I would appreciate an explanation of why the paper is written as if presenting tabular Q-learning for TNDP and MNEP, instead of explaining that often TNDP and MNEP problems are small enough to allow for tabular Q-learning to be efficient but as or more accurate than more computationally costly DeepRL.
>
> **Response:** Please note that previous approaches applied Deep RL without further reflecting on the proper scale of the problem; as the reviewer correctly highlights, our point is that tabular methods suffice for TNDP/MNEP problems with the granularity (i.e., network size and neighborhood size) of real-world problems. We have adapted the introduction to make it clearer that tabular methods can work well, since the MNEP has a limited set of features and is constrained in scale. We now explicitly mention that as the reason why we select a tabular Q-learning approach.
>
> Thank you again for the constructive review and for helping us improve the paper!

---

### Review · Reviewer_Tfdm · 2025-02-09

**Summary Of Contributions:**

This paper attempts to reformulate the Metro Network Expansion Problem (MNEP) as an MDP with finite state space and a small bounded and fixed action space, such that  tabular reinforcement learning can be applied. The paper shows that a Q-value iteration method achieves similar performance as previous methods using deep RL or other heuristics, using an order of magnitude fewer training episodes.

**Audience:**

No

**Claims And Evidence:**

No

**Requested Changes:**

Please fix the claims regarding points 1, 2, and 4 above, and please reconsider the formulation regarding point 3.

On another note: while I have no objections to application papers, I am unable to see how this paper fits any of the 9 categories of invited papers listed here: https://jmlr.org/tmlr/editorial-policies.html. I do not think this paper falls under "formalization of new learning tasks (e.g., in the context of new applications) and of methods for assessing performance on those tasks;", since MNEP is not a new learning task. Nor does it fall under "accounts of applications of existing techniques that shed light on the strengths and weaknesses of the methods;" since the strengths/weaknesses of tabular versus deep RL are well known.

**Strengths And Weaknesses:**

Strengths:
* It is good to show that deep RL is not strictly necessary when simpler and more interpretable methods perform just as well.

Weaknesses:
There are some critical errors and unsupported claims in this paper:
1. Interpretability: this paper claims that the proposed tabular RL method produces policies that are more interpretable than those of deep RL. However, the evidence provided in Section 6.3 do not justify this claim. Figure 6a shows a transport line generated by the resulting policy - this is just the result of running any policy, whether deep or tabular. Figure 6b shows the average reward from different starting points - this plot can also be generated by computing the total reward from running a deep policy from the starting points. Figure 6c shows the Q-values of each cell - any deep RL method with a value function can also generate this plot. Hence the paper's claim does not hold.
2. "Balancing efficiency with fairness": this paper claims that the proposed method balances efficiency with fairness. However, all I can see are three separate reward functions: the pure efficiency reward in equation (1), and the equal-sharing reward equation (3) and Rawls' reward equation (4). It is not as though the proposed tabular RL method is some kind of multi-objective RL method that optimizes the Pareto front of efficiency and equity. Simply experimeting with different rewards separately does not mean that the *method* balances competing objectives.
3. Critical error: Section 4.2 says that the reward for an action (adding a new station) depends on all previously selected stations on the new metro line. However, the state $s$ used for the tabular $Q(s,a)$ only contains the current location, not the locations of the previous selected stations. This means the $Q(s,a)$is unable to differentiate between two completely different scenarios where the existing lines are different, if the $s$ input is the same cell.
4. The paper calls the proposed method a Q-learning method, but the update rule used in equation (6) is not Q-learning (which needs the max operator).

Other minor errors:
* Intro paragraph "by maximizing maximize"
* Section 3.1 has a duplicate paragraph "Each cell..."
* Equation (3) has an extra comma between $W_i,U...$
* Second line of page 6: non-increasing weights should mean $W_1 \geq W_2 \geq ...$.

---

> ### Author Response · Authors · 2025-02-20
> **Reponse to Reviewer Tfdm (Part 1)**
>
> We thank the reviewer for taking the time to carefully read our paper and provide very constructive comments. We are extremely pleased to read that reviewer Tfdm recognizes the value of our contribution in showcasing that Deep RL is not strictly necessary when simpler and more interpretable methods perform just as well. Please find our point-by-point response below.
>
> **Comment:** This paper claims that the proposed tabular RL method produces policies that are more interpretable than those of deep RL. However, the evidence provided in Section 6.3 does not justify this claim. Figure 6a shows a transport line generated by the resulting policy—this is just the result of running any policy, whether deep or tabular. Figure 6b shows the average reward from different starting points—this plot can also be generated by computing the total reward from running a deep policy from the starting points. Figure 6c shows the Q-values of each cell—any deep RL method with a value function can also generate this plot. Hence, the paper's claim does not hold.
>
> **Response:** We acknowledge the concerns that similar visualizations could be produced using deep RL. However, our claim regarding interpretability is not solely about the ability to generate visual representations, but rather about how values are structured and accessed. In our method, each state-action pair is explicitly stored, making the learned policy fully transparent. This differs from deep RL, where extracting a clear mapping from states to actions requires additional methods. While deep RL can approximate Q-values, the challenge lies in interpreting how these values are derived, particularly for stakeholders in urban planning who require explicit decision rationales. Furthermore, please note that the visualizations present in Figure 6 (Figure 5 in the latest version) are only possible given the state-action formulation we originally propose to solve the TNDP, where the state is the location of the agent (see section 4.1). With the previous representation, this level of state inspection is not possible. We have updated Section 6.3 to emphasize this, and it now also mentions that these plots could be produced in deep RL settings too.
>
> **Comment:** This paper claims that the proposed method balances efficiency with fairness. However, all I can see are three separate reward functions: the pure efficiency reward in Equation (1), and the equal-sharing reward Equation (3) and Rawls' reward Equation (4). It is not as though the proposed tabular RL method is some kind of multi-objective RL method that optimizes the Pareto front of efficiency and equity. Simply experimenting with different rewards separately does not mean that the method balances competing objectives.
>
> **Response:** We agree that the formulation of this claim is not fully accurate, and we have reworded it to indicate that the reward functions can be used to achieve both efficiency and fairness, refraining from using the word "balancing," which could imply that we demonstrate both objectives being optimized simultaneously.
>
> **Comment:** Critical error: Section 4.2 says that the reward for an action (adding a new station) depends on all previously selected stations on the new metro line. However, the state used for the tabular only contains the current location, not the locations of the previous selected stations. This means the is unable to differentiate between two completely different scenarios where the existing lines are different, if the input is the same cell.
>
> **Response:** Thank you for pointing this out. We have revamped section 4.1 to clarify our formulation, which indeed is an MDP with non-markovian rewards. We also further updated 4.1 to add more discussion on this choice.
>
> **Comment:** Section 4.2 says that the reward for an action (adding a new station) depends on all previously selected stations on the new metro line. However, the state used for the tabular method only contains the current location, not the locations of the previously selected stations. This means the model is unable to differentiate between two completely different scenarios where the existing lines are different if the input is the same cell.
>
> **Response:** We have revamped Section 4.1 to clarify our formulation, which indeed represents an MDP with non-Markovian rewards. We have also further updated Section 4.1 to add more discussion on this choice, mentioning why we opted for the Non-markovian rewards and citing appropriate literature.
>
> **Comment:** The paper calls the proposed method a Q-learning method, but the update rule used in Equation (6) is not Q-learning (which requires the max operator).
>
> **Response:** The update rule used in Equation (6) is based on Monte Carlo estimation, where the agent updates the Q-values using the total discounted return from the episode. This is a key difference from standard Q-learning, which uses the max operator.

---

> > ### Author Response · Authors · 2025-02-20
> > **Reponse to Reviewer Tfdm (Part 2)**
> >
> > **Comment:** Minor errors: Introduction paragraph contains "by maximizing maximize"; Section 3.1 has a duplicate paragraph starting with "Each cell..."; Equation (3) has an extra comma; Second line of page 6: "non-increasing weights" should be corrected.
> >
> > **Response:** We have fixed all these issues and changed the wording to "strictly decreasing," which more accurately captures the reward function.
> >
> > **Comment:** While I have no objections to application papers, I am unable to see how this paper fits any of the nine categories of invited papers listed here: https://jmlr.org/tmlr/editorial-policies.html. I do not think this paper falls under "formalization of new learning tasks (e.g., in the context of new applications) and of methods for assessing performance on those tasks," since MNEP is not a new learning task. Nor does it fall under "accounts of applications of existing techniques that shed light on the strengths and weaknesses of the methods," since the strengths/weaknesses of tabular versus deep RL are well known.
> >
> > **Response:**
> > We understand your concern about how our work fits into TMLR. We would like to provide some explanation to clarify how our research contributes to the journal's topics.
> >
> > 1. **Applications of existing techniques to reveal strengths and weaknesses:** While the strengths and weaknesses of Deep RL and tabular methods are well studied in general, it remains unclear when, in practice, each should or should not be used. The Metro Network Expansion Problem (MNEP) is a prime example of a problem that transitioned from traditional optimization methods to Deep RL without careful evaluation. By analyzing both performance and emissions, we highlight the limitations of Deep RL for combinatorial optimization problems with small feature sets. This issue is often overlooked, as recent works tend to adopt Deep Learning methods without first considering more traditional approaches.
> >
> > 2. **Formalization of new learning tasks:** While MNEP itself is not a new problem, our approach—reformulating learning into two stages (initial placement and local movement)—is novel and has not been explored before. Additionally, we demonstrate that non-Markovian rewards can achieve performance comparable to Deep RL, even when relaxing the theoretical guarantees of MDPs. Moreover, while not the main contribution of our paper, we also use a model to estimate OD matrices of Amsterdam, extending a method previously applied to Xi'an in a new setting where mobility data is not available, but needs to be inferred.
> >
> > We believe our research aligns with TMLR’s scope and contributes to the broader understanding of how AI methods can be better used in real-world policy and urban planning settings. We welcome any further feedback or clarification on how we can better align our work with the journal’s objectives.
> >
> > We thank you again for the feedback, which helped us improve the paper significantly.

---

> > > ### Comment · Reviewer_Tfdm · 2025-03-02
> > > **Acknowledgement of reviewer response, part 2**
> > >
> > > > Applications of existing techniques to reveal strengths and weaknesses
> > >
> > > The issue is that the conclusions about tabular versus deep RL in this paper are either already known in general (one can just run tabular and deep RL on any small finite state-action environment) or are specific to this MNEP problem, and hence I am uncertain about relevance to TMLR readers who do not work on this application.
> > >
> > > > Formalization of new learning tasks: While MNEP itself is not a new problem, our approach—reformulating learning into two stages (initial placement and local movement)—is novel and has not been explored before.
> > >
> > > I can see your point that it is somewhat a new learning task because the MDP formulation is different from the previous work on the same application. Depends on whether one takes a strict interpretation of "Formalization of new learning tasks". My initial expectation is that it must be a completely new application and you need to be the first to formulate it as an RL problem.

---

> > > > ### Author Response · Authors · 2025-03-05
> > > > **Response to further comments by Reviewer Tfdm**
> > > >
> > > > We would like to thank you again. Your contributions have been immensely valuable.
> > > >
> > > > > Given a finite state-action space, deep networks have the same "transparency" as tabular methods. It's just a Q-function and/or a policy. Unless you are extracting symbolic expressions for the policy after running RL (there are methods for doing so), the answer to "How are these values derived" is "we used RL", and it does not matter whether the method is tabular or neural net representations since the state-action space is finite and not hard to enumerate to characterize the Q-function/policy for cases that someone may want to inspect.
> > > >
> > > > We acknowledge that deep RL methods can theoretically enumerate Q-values. However, the interpretability of our method is not only how Q-values are derived, but also how easy it is to read and interpret them. In contexts where communication with different stakeholders is necessary (such as urban planning) we argue that enumerating values is not sufficient to interpret and convey a solution. Given the large number of layers and non-linear transformations involved in deriving q-values for DeepRL, explaining the advantages of a solution would require extra details about function approximation. What DeepRL methods learn are often complex and require posthoc methods to extract meaningful representations. Our approach does not suffer from this extra step, but offers direct, human-readable access to the values that support a decision, which is particularly relevant in urban planning contexts where stakeholders may not be familiar with function approximators and ANNs. Note that we are not claiming tabular RL is more interpretable in any problem, but for our specific MNEP formulation. Furthermore, please note that interpretability, in our case, also stems from reducing the problem to a smaller state-space. We thank you for this discussion, that will help us to re-formulate the meaning of interpretability in our paper.
> > > >
> > > > >Good to see this correction in the writing, but it makes one wonder whether better solutions can be found by a formulation where the current full design is captured by the state definition.
> > > >
> > > > In fact, considering the full design is how the problem has been modeled so far, which explains the large state-space. In our formulation, the state-space is significantly reduced, leading to notable improvements in computational efficiency while maintaining solution quality comparable to previous approaches.
> > > > Thank you for your comment—it highlights the need to emphasize that our formulation produces results consistent with those obtained when with the full state-space. We will make this connection explicit in our paper.
> > > >
> > > > >Then you are doing value iteration, and the term "Q-learning" should not appear anywhere in the paper. e.g. "We propose a tabular Q-learning algorithm" makes me expect a max operator.
> > > >
> > > > We appreciate the comment and indeed, to avoid confusion we will stop referring to the method as Q-learning.
> > > >
> > > > >The issue is that the conclusions about tabular versus deep RL in this paper are either already known in general (one can just run tabular and deep RL on any small finite state-action environment) or are specific to this MNEP problem, and hence I am uncertain about relevance to TMLR readers who do not work on this application.
> > > >
> > > > This is a good observation and we understand your concern. However, our work provides a concrete case study demonstrating that deep RL is not always necessary, even in problems where it has become a common approach. While this may not introduce new theoretical insights about RL methods themselves, it concerns the practical application of RL in real-world scenarios, a topic still under heavy research. As we mention in the paper, we shed light to other applications of combinatorial optimization that could benefit from such reformulation, and thus is relevant to TMLR readers.
> > > >
> > > > >I can see your point that it is somewhat a new learning task because the MDP formulation is different from the previous work on the same application. Depends on whether one takes a strict interpretation of "Formalization of new learning tasks". My initial expectation is that it must be a completely new application and you need to be the first to formulate it as an RL problem.
> > > >
> > > > We appreciate your perspective and acknowledge that our work is not the first to formalize MNEP as an RL problem. However, we argue that this does not mean it does not constitute a new learning task. In contrast, We believe there is immense value in revisiting and refining problem formulations to advance our understanding, particularly when proposing a new formulation that ultimately leads to solutions with similar performance yet requiring a much less complex method (both computationally and conceptually). This is particularly relevant for MNEP, where RL-based approaches are still new and developing, and no widely accepted standard formulation exists. By this new formulation, we contribute to this ongoing exploration.

---

> > ### Comment · Reviewer_Tfdm · 2025-03-02
> > **Acknowledgement of reviewer response part 1**
> >
> > Regarding interpretability:
> > > In our method, each state-action pair is explicitly stored, making the learned policy fully transparent. This differs from deep RL, where extracting a clear mapping from states to actions requires additional methods. While deep RL can approximate Q-values, the challenge lies in interpreting how these values are derived, particularly for stakeholders in urban planning who require explicit decision rationales.
> >
> > Given a finite state-action space, deep networks have the same "transparency" as tabular methods. It's just a Q-function and/or a policy. Unless you are extracting symbolic expressions for the policy after running RL (there are methods for doing so), the answer to "How are these values derived" is "we used RL", and it does not matter whether the method is tabular or neural net representations since the state-action space is finite and not hard to enumerate to characterize the Q-function/policy for cases that someone may want to inspect.
> >
> > > we have reworded it to indicate that the reward functions can be used to achieve both efficiency and fairness, refraining from using the word "balancing," which could imply that we demonstrate both objectives being optimized simultaneously.
> >
> > Good to see this clarification.
> >
> > > We have revamped section 4.1 to clarify our formulation, which indeed is an MDP with non-markovian rewards. We also further updated 4.1 to add more discussion on this choice.
> >
> > Good to see this correction in the writing, but it makes one wonder whether better solutions can be found by a formulation where the current full design is captured by the state definition.
> >
> > > The update rule used in Equation (6) is based on Monte Carlo estimation, where the agent updates the Q-values using the total discounted return from the episode. This is a key difference from standard Q-learning, which uses the max operator.
> >
> > Then you are doing value iteration, and the term "Q-learning" should not appear anywhere in the paper. e.g. "We propose a tabular Q-learning algorithm" makes me expect a max operator.

---

### Review · Reviewer_41vj · 2025-02-09

**Summary Of Contributions:**

This paper proposes a tabular reinforcement learning (RL) algorithm for the Metro Network Expansion Problem. After introducing the algorithm, numerical experiments on two environments are reported. These experiments correspond to two different cities. In both cases, the results show that the RL algorithm converges faster than a reference deep RL method.

**Audience:**

Yes

**Broader Impact Concerns:**

There could be various ethical implications if cities were really using such algorithms for metro planning. Please add a paragraph about broader impact.

**Claims And Evidence:**

No

**Requested Changes:**

Please provide more explanations about why the tabular method performs better than the deep RL method. In particular, please explain carefully the number of state and actions, and provide a sweep of hyperparameters of the deep RL method to show that you used the best hyperparameter values.

Please provide more details on the question of interpretability, to support your claim in Section 6.3.

**Strengths And Weaknesses:**

Strengths: (1) The class of problems tackled in this paper is an important one for real-world applications. (2) It is interesting that tabular RL performs well on the two examples considered here.

Weaknesses: (1) Aside from the numerical results, there is little explanation for why the tabular method should be more efficient than the deep RL method. One possibility is that the problems considered here are too small for deep RL to benefit from the scalability offered by neural networks. Alternatively, the deep RL algorithm's hyperparameters may not have been carefully tuned, resulting in poor performance. (2) Another issue is the claim that the policies are more 'interpretable' (Section 6.3), which is unclear. While a tabular method directly represents the policy state by state, whereas a deep neural network requires evaluation at each state, this does not necessarily imply 'interpretability' in terms of providing a clear explanation for the chosen actions.

---

> ### Author Response · Authors · 2025-02-20
> **Response to Reviewer 41vj (Part 1)**
>
> We thank reviewer 41vj for taking the time to provide a valuable and helpful review. We are very happy to read that you recognize the importance of the real-world application we are focusing on, and the value of tabular RL methods. Please find below our point by point response and overview of changes.
>
> **Comment:** Aside from the numerical results, there is little explanation for why the tabular method should be more efficient than the deep RL method. One possibility is that the problems considered here are too small for deep RL to benefit from the scalability offered by neural networks.
>
> **Response** We have updated both the introduction and the conclusion to better reflect our justification. Indeed, we argue that although the problems we tackle involve complex solution spaces, they are fundamentally static optimization problems with limited input features. Their scalability is further constrained due to distance and shape constraints. This makes neural networks not necessarily the ideal choice of algorithm.
>
> **Comment:** Alternatively, the deep RL algorithm's hyperparameters may not have been carefully tuned, resulting in poor performance.
>
> **Response** We used the hyperparameters reported by the original authors of each of the baselines --- for the Genetic Algorithm, where there are two existing sets of hyperparameters in previous work, we evaluated both and used the ones with the better performance. We have added a new section on the Appendix detailing our hyperparameter selection methodology (Appendix C).
>
> **Comment:** Another issue is the claim that the policies are more 'interpretable' (Section 6.3), which is unclear. While a tabular method directly represents the policy state by state, whereas a deep neural network requires evaluation at each state, this does not necessarily imply 'interpretability' in terms of providing a clear explanation for the chosen actions. Please provide more details on the question of interpretability, to support your claim in Section 6.3.
>
> **Response**
> As we mentioned in our response to reviewer FpTH, while Deep RL methods can, in theory, extract Q-values, their interpretability differs from our approach. Neural networks do not store explicit state-action value mappings in an accessible way; instead, they encode them in high-dimensional latent spaces, making it hard to understand how specific inputs influence decisions. Extracting this information requires post-hoc analysis for each state-action pair, typically through saliency maps or t-SNE plots, which are sensitive to simple transformations [1] and generally unreliable [2]. In our approach, instead, all that is required is imply to store the Q-table after training.
>
> Furthermore, in deep Q-networks, Q-values are often estimated using probability distributions over actions, requiring additional steps to devise a single, interpretable value. In contrast, TabularMNEP maintains an explicit tabular representation of the best estimation of the current Q-values, allowing for direct inspection and interpretation. The ability to directly query state-action values enables straightforward policy evaluation, even by non-experts, which is not as readily feasible with Deep RL.
>
> Furthermore, our method inherently allows one to identify underexplored areas of the state space by observing Q-value distributions, which is significantly harder with deep models. This offers flexibility in the designer/decision-maker for further post-training adjustments.
>
> [1] Kindermans, P. J., Hooker, S., Adebayo, J., Alber, M., Schütt, K. T., Dähne, S., ... \& Kim, B. (2019). The (un) reliability of saliency methods. Explainable AI: Interpreting, explaining and visualizing deep learning, 267-280.
>
> [2] Annasamy, R. M., \& Sycara, K. (2019, July). Towards better interpretability in deep q-networks. In Proceedings of the AAAI conference on artificial intelligence (Vol. 33, No. 01, pp. 4561-4569).
>
> While deep RL can approximate Q-values, the challenge lies in interpreting how these values are derived, particularly for stakeholders in urban planning who require explicit decision rationales. We have updated Section 6.3 to emphasize this.

---

> > ### Author Response · Authors · 2025-02-20
> > **Response to Reviewer 41vj (Part 2)**
> >
> > **Requested Change:** Please provide more explanations about why the tabular method performs better than the deep RL method. In particular, please explain carefully the number of state and actions, and provide a sweep of hyperparameters of the deep RL method to show that you used the best hyperparameter values.
> >
> > **Response** As mentioned above, we have added an additional section in the Appendix (Appendix C) in which we report, in detail, the hyperparameters we used, together with a description of the process of selected them.
> >
> > **Requested Change:** Please provide more details on the question of interpretability, to support your claim in Section 6.3.
> >
> > **Response** We have updated Section 6.3 to emphasize the advantage of inherent interpretability in TabularMNEP compared to the DeepRL alternative.
> >
> > **Broader Impact Response:** There could be various ethical implications if cities were really using such algorithms for metro planning. Please add a paragraph about broader impact.
> >
> > **Response** Thank you for this suggestion. We have added a paragraph in the conclusion in which we address the broader impact and call for a careful consideration of applying reinforcement learning to the metro network design problem.
> >
> > Thank you again for your valuable feedback, it helped us improve the paper significantly!

---

### Review · Reviewer_Amvs · 2025-02-18

**Summary Of Contributions:**

[I would like to apologize for my late review. I was sick and failed to inform the journal in time. I took care not to read the other reviewers comments before laying out my own thoughts]

The paper compares tabular and neural network-based implementations of reinforcement learning in a transportation network problem. It shows that tabular approaches can be sufficient and perform on par with deep learning based approaches

**Audience:**

No

**Claims And Evidence:**

No

**Requested Changes:**

See above

**Strengths And Weaknesses:**

# Strengths
The paper considers a broad setup of questions on their chosen benchmark (transportation planning in various cities, with various reward functions / utilities) and provides a good experimental setup to compare different approaches.
The authors focus on often neglected aspects of computational efficiency and finding the smallest solution to a given problem.

# Conceptual issues
* **Problem vs solution engineering** My first major issue with the paper is the question whether the baseline method and the tabular method solve the _same_ problem. In this paper, the core claim is about the efficiency of tabular methods. However, in the experiment design, the authors propose changes to the action space. This is an orthogonal choice to the solution method and I am not fully sure if the impacts were disentangled in a relevant way. Given the text, I interpret that the authors compared their tabular method with a significantly smaller action space against prior work's deep learning approach on a large action space (every cell). This however seems like an unequal comparison to me, as it is feasible to run a deep learning based approach on the modified problem as well.
* **Unclear relevance of fairness discussion** While I do not doubt that fairness is a relevant criterion for planning transportation in the real world, the discussion presented in the paper seems orthogonal to the technical question to me. Both tabular RL methods and deep RL ones optimize the same, single reward criterion in their standard formulation. The way that fairness is presented here is through differing reward criteria. It is unclear to me whether the authors are trying to make a claim of the form "Rewards that are obtained from e.g. Rawl's theory of justice have properties that make them better suited to tabular methods", and if this is indeed the claim, I would like to see much more theoretical and empirical evidence for this claim. While I applaud the authors efforts to verify the efficacy of their approach with different reward criteria, the limited variability of performance across these suggests, in my eyes, that the technical contribution is indeed orthogonal to the choice of reward.
* **Limited generalisability of insights** My final major issue is the scope of the research question. Tabular and Deep RL have different strengths, and a purely empirical paper is in my opinion insufficient to compare both. The implication here is that transportation planning is better suited to tabular RL. However, the qualitative claim here would in my opinion be better replaced by an quantitative claim: up to what level of problem complexity can we expect tabular methods to outperform neural network-based ones? As Deep RLs main promise is scaling (which the authors themselves point out) this seems like an important question. I am not a transportation domain expert and therefore it is very hard for me to judge whether the environments considered by the authors are "the real deal" i.e. could be used for actual decision making in the wild, or whether they are more akin to toy problems. But putting on my hat as an RL practitioner, reading this paper I ask myself: how can I tell if my problem is similar enough to the one presented here, so that the claims made apply to my problem as well.

# Methodological issues
* **CO2 measurements** The CO2 measurements are not well described and the authors themselves acknowledge flaws in the setup as they use GPU nodes without using the GPUs. I think claims of computational efficiency are sufficient on their own, and carbon emissions can be used as a motivation why readers should care about this. But as a quantitative measurement in the paper, I do not think the comparison is done with sufficient care.
* **Interpretability** The authors show the interpret-ability of their setup using a 2d grid visualization of values. However, as pointed out above, neural networks can operate on the same input and the same visualization can therefore be presented for these. The interpretability of the results as presented here seems to be a feature of the problem, not of the model.
* **Baseline construction** As outlined above, it is unclear on what problem the baseline (deep RL) was executed. In addition, the paper has no information on any technical details: implementation, hyperparameters, methods for tuning these for each environment and reward criterion. As such, the paper cannot be evaluated for correctness in its current state.


# Minor issues
* Please use appropriate parentheses for citations.

---

> ### Author Response · Authors · 2025-02-20
> **Response to Reviewer Amvs (Part 1)**
>
> We would like to thank reviewer Amvs for taking the time to provide a valuable and helpful review, and wish you a speedy recovery. Please find below our point by point response and overview of changes.
>
> **Conceptual Issues**
>
> **Comment:** My first major issue with the paper is the question whether the baseline method and the tabular method solve the same problem. In this paper, the core claim is about the efficiency of tabular methods. However, in the experiment design, the authors propose changes to the action space. This is an orthogonal choice to the solution method and I am not fully sure if the impacts were disentangled in a relevant way. Given the text, I interpret that the authors compared their tabular method with a significantly smaller action space against prior work's deep learning approach on a large action space (every cell). This however seems like an unequal comparison to me, as it is feasible to run a deep learning based approach on the modified problem as well.
>
> **Response:** Thank you for your thoughtful comment. We confirm that both the baseline methods and our proposed tabular method address the same problem: the combinatorial optimization problem of Metro Network Expansion (MNEP). The key difference lies in our reformulation of the problem, which significantly reduces the state and action spaces.
>
> Our primary claim is that many combinatorial optimization problems, which are increasingly approached with Deep Learning methods, can instead be tackled more efficiently through careful problem reformulation. This reformulation is a fundamental part of our contribution, as it enables more tractable solutions without necessarily relying on deep learning.
>
> Importantly, and close  to your own thought, our reformulated problem remains compatible with Deep RL methods. For instance, one could leverage Deep Learning for only the first stage of the problem --- selecting the initial expansion cell, while still benefiting from our reformulation. This demonstrates that our method does not inherently disadvantage deep learning approaches, but rather provides a more efficient problem framing that could be applied broadly to other combinatorial optimization problems.
>
> By advocating for using the most appropriate method for each problem formulation, we contribute to improvements in method usage by non-experts, efficiency, interpretability, and energy efficiency. We hope this clarification addresses your concern and illustrates the broader implications of our approach.
>
> **Comment:** Unclear relevance of fairness discussion. While I do not doubt that fairness is a relevant criterion for planning transportation in the real world, the discussion presented in the paper seems orthogonal to the technical question to me. Both tabular RL methods and deep RL ones optimize the same, single reward criterion in their standard formulation. The way that fairness is presented here is through differing reward criteria. It is unclear to me whether the authors are trying to make a claim of the form "Rewards that are obtained from e.g. Rawl's theory of justice have properties that make them better suited to tabular methods", and if this is indeed the claim, I would like to see much more theoretical and empirical evidence for this claim. While I applaud the authors efforts to verify the efficacy of their approach with different reward criteria, the limited variability of performance across these suggests, in my eyes, that the technical contribution is indeed orthogonal to the choice of reward.
>
> **Response:**
> We appreciate your insightful comment. To clarify, we do not claim that social-good-based reward functions possess properties that make them inherently better suited for tabular methods.
>
> Our primary motivation for incorporating fairness-based rewards is twofold. First, we aim to bridge the gap between machine learning and transport planning by integrating objectives commonly used in transport planning research. Second, as you noted, we seek to demonstrate the generality of our method by showing its ability to optimize different types of reward functions, rather than being effective only under a single, arbitrary, efficiency-based criterion.
>
> Furthermore, introducing fairness-based rewards increases the complexity of the problem. Unlike the Maximum Efficiency reward, which operates on a single OD matrix of size  $n \times m$ , fairness-based rewards require a separate OD matrix for each group, resulting in a final OD matrix of size  $5 \times n \times m$. This added complexity ensures that our method is tested under more realistic and diverse conditions.
>
> In summary, our goal is to align the problem more closely with real-world transport planning challenges while also demonstrating that our method’s performance is robust across different reward formulations. We hope this clarifies our motivation and addresses your concern.

---

> > ### Author Response · Authors · 2025-02-20
> > **Response to Reviewer Amvs (Part 2)**
> >
> > **Comment:** Limited generalisability of insights. My final major issue is the scope of the research question. Tabular and Deep RL have different strengths, and a purely empirical paper is in my opinion insufficient to compare both. The implication here is that transportation planning is better suited to tabular RL. However, the qualitative claim here would in my opinion be better replaced by an quantitative claim: up to what level of problem complexity can we expect tabular methods to outperform neural network-based ones? As Deep RLs main promise is scaling (which the authors themselves point out) this seems like an important question. I am not a transportation domain expert and therefore it is very hard for me to judge whether the environments considered by the authors are "the real deal" i.e. could be used for actual decision making in the wild, or whether they are more akin to toy problems. But putting on my hat as an RL practitioner, reading this paper I ask myself: how can I tell if my problem is similar enough to the one presented here, so that the claims made apply to my problem as well.
> >
> > **Response:** Thank you for your thoughtful comment. You are absolutely right that our original submission did not provide enough clarity on this point. To address this, we have now updated our paper to include a more detailed discussion.
> >
> > Specifically, we clarify in the introduction that metro network expansion (MNEP) problems inherently have low-dimensional feature spaces and natural constraints on their scalability. For example, in real-world settings, metro stations are typically placed between $1–-3$ km apart. In our experiments in Amsterdam, we deliberately increased the granularity by placing stations $0.5$ km apart, effectively pushing the limits of the problem’s complexity. This reinforces our argument that MNEP is not a problem that scales indefinitely, making it well-suited for simpler methods like tabular RL.
> >
> > Additionally, we have revamped Section 4.1, where we reformulate the problem, to provide more details on its time complexity and scale. We have also expanded our discussion of the main components of the Markov Decision Process (MDP) to offer a clearer justification for our methodological choices.
> >
> > Regarding your final point---how RL practitioners can determine whether our insights apply to their own problems---this is precisely one of our motivations for this research. RL is a powerful tool for sequential decision-making, but our work demonstrates that careful problem formulation can often make complex Deep RL models unnecessary. In some cases, rethinking the structure of a combinatorial optimization problem can allow traditional, more interpretable methods to perform just as well as deep learning approaches.
> >
> > We hope these clarifications help address your concerns, and we appreciate your valuable feedback.
> >
> > **Methodological issues**
> >
> > **Comment:** CO2 measurements The CO2 measurements are not well described and the authors themselves acknowledge flaws in the setup as they use GPU nodes without using the GPUs. I think claims of computational efficiency are sufficient on their own, and carbon emissions can be used as a motivation why readers should care about this. But as a quantitative measurement in the paper, I do not think the comparison is done with sufficient care.
> >
> > **Response:** You are correct, but we note that accurately measuring the total carbon emissions of training Machine Learning models remains an open challenge, as there is no universally accepted methodology. While some popular tools, such as the \textit{experiment-impact-tracker} library, exist, they are often outdated and no longer maintained.
> >
> > In our paper, we rely on the approach described in ML CO2 Impact [1], which is one of the more widely used methods for estimating emissions. We acknowledge that a more precise measurement would involve fully decoupling GPU and CPU usage. However, this distinction does not affect our central claim---if anything, it would further strengthen our argument about the energy efficiency of our approach. We have added the appropriate citation in our text.
> >
> > [1] Lacoste, A., Luccioni, A., Schmidt, V., \& Dandres, T. (2019). Quantifying the carbon emissions of machine learning. arXiv preprint arXiv:1910.09700.

---

> > > ### Author Response · Authors · 2025-02-20
> > > **Response to Reviewer Amvs (Part 3)**
> > >
> > > **Comment:** Interpretability The authors show the interpretability of their setup using a 2d grid visualization of values. However, as pointed out above, neural networks can operate on the same input and the same visualization can therefore be presented for these. The interpretability of the results as presented here seems to be a feature of the problem, not of the model.
> > >
> > > **Response:** Our interpretability claim is based on two features of our paper: the reformulation of the problem to allow for representing grid cells as states, and the usage of tabular RL.
> > >
> > > We acknowledge that the problem tackled here presents a state-space that is amenable to a visual representation in a 2D grid. However, it is important to note that previous works did not explore the possibility of representing Figure 6. In fact, this visualization (Figure 5 in the revised version of the paper) is only possible due to our novel state-action formulation for solving the TNDP, where the state corresponds to the agent’s location (see Section 4.1). Under previous representations, this level of state inspection was not achievable.
> > >
> > > As we mentioned in our response to reviewer FpTH, while Deep RL methods can, in theory, extract Q-values, their interpretability differs from our approach. Neural networks do not store explicit state-action value mappings in an accessible way; instead, they encode them in high-dimensional latent spaces, making it hard to understand how specific inputs influence decisions. Extracting this information requires post-hoc analysis for each state-action pair, typically through saliency maps or t-SNE plots, which are sensitive to simple transformations [1] and generally unreliable [2]. In our approach, instead, all that is required is imply to store the Q-table after training.
> > >
> > > Furthermore, in deep Q-networks, Q-values are often estimated using probability distributions over actions, requiring additional steps to devise a single, interpretable value. In contrast, TabularMNEP maintains an explicit tabular representation of the best estimation of the current Q-values, allowing for direct inspection and interpretation. The ability to directly query state-action values enables straightforward policy evaluation, even by non-experts, which is not as readily feasible with Deep RL.
> > >
> > > Furthermore, our method inherently allows one to identify underexplored areas of the state space by observing Q-value distributions, which is significantly harder with deep models. This offers flexibility in the designer/decision-maker for further post-training adjustments.
> > >
> > >
> > > [1] Kindermans, P. J., Hooker, S., Adebayo, J., Alber, M., Schütt, K. T., Dähne, S., ... \& Kim, B. (2019). The (un) reliability of saliency methods. Explainable AI: Interpreting, explaining and visualizing deep learning, 267-280.
> > >
> > > [2] Annasamy, R. M., \& Sycara, K. (2019, July). Towards better interpretability in deep q-networks. In Proceedings of the AAAI conference on artificial intelligence (Vol. 33, No. 01, pp. 4561-4569).
> > >
> > > **Comment:** Baseline construction As outlined above, it is unclear on what problem the baseline (deep RL) was executed. In addition, the paper has no information on any technical details: implementation, hyperparameters, methods for tuning these for each environment and reward criterion. As such, the paper cannot be evaluated for correctness in its current state.
> > >
> > > **Response:** We have now added a new Appendix section (Appendix C) that gives details on the hyperparameter search we used for our method, as well as the origins of the baselines and the hyperparameters used for them.
> > >
> > > **Minor issues**
> > >
> > > **Comment:** Please use appropriate parentheses for citations.
> > >
> > > **Response:** We have updated our citations throughout the text to include parentheses.

---

### Author Response · Authors · 2025-02-20
**Response to all Reviewers**

Dear Reviewers,

We sincerely appreciate your thorough, thoughtful, and constructive feedback on our paper.

We are grateful that many of you acknowledge the strengths of our work and recognize our contribution in reframing the Metro Network Expansion problem to allow for simpler methods. We also appreciate your recognition of our proposed tabular RL method and its efficiency in addressing this challenge.

We understand and acknowledge the concerns raised by some of you regarding interpretability. Please refer to our response and the updated paper for clarification. We argue that interpretability emerges from a combination of both the problem reframing and state-space reduction, as well as the use of a tabular RL method. We kindly ask you to consider the reframing of the problem as a key contribution of our work in your evaluation.

We have carefully considered all of your comments and suggestions for improvement, leading to several revisions in the paper. We believe these changes effectively address the concerns raised in your reviews and enhance the overall quality of the manuscript.

We provide an annotated version of the paper, with the changes highlighted in blue color. In our answers to your comments, we refer to these changes for easier retrieval.

Once again, we sincerely thank you for your time and effort in reviewing our paper and for your valuable feedback.

---

### Decision · Action_Editor_V8Qg · 2025-03-17

**Recommendation:** Reject

**Comment:**

The paper mainly contributes in tabular RL algorithms. The concerns raised by the reviewers are critical and need to be addressed with a resubmission. These include limited scalability and unclear writing. Although the authors made significant efforts to clarify during the rebuttal period, I still hope the authors can substantially revise the paper to more thoroughly address these issues.

Some of the suggested edits for the final version:

Methodology and Comparison Fairness:
- The comparison between tabular and deep RL methods appears to be conducted under different action space configurations, making it difficult to draw valid conclusions about their relative performance.
- The experimental setup for CO2 measurements and computational efficiency comparisons requires more rigorous methodology.

Limited Scope and Generalizability:
- The paper lacks sufficient theoretical foundation to support its claims about the superiority of tabular methods.
- There is insufficient analysis of the problem complexity threshold where tabular methods might outperform neural network-based approaches.

Technical Clarity:
- The implementation details, hyperparameter settings, and tuning methodology for the deep RL baseline are inadequately described.
- The claims about interpretability need stronger justification, as the visualizations presented appear to be more a feature of the problem rather than the method.

Fairness Discussion:
- The connection between the technical contribution and the fairness considerations needs to be better established and justified.

**Audience:**

The reviewers acknowledged the effectiveness and ease of application of this work. However, they also pointed out that the submission demonstrates improved presentation and soundness but lacks enough comparison with necessary baselines. More importantly, reviewers mentioned that the experimental results are not convincing, and that TMLR's audience who do not work on this application will not benefit from reading this work. The main conclusion, that tabular RL is more efficient for small MDPs, is deemed insufficient for publication. Additionally, the non-Markovian formulation is questionable.

**Claims And Evidence:**

This paper proposes a tabular reinforcement learning (RL) algorithm for the Metro Network Expansion Problem. After careful consideration of the reviewer feedback, I regret to inform the authors that their manuscript cannot be accepted for publication in its current form. While the reviewers acknowledge the paper's strengths in addressing an important real-world application and demonstrating the potential effectiveness of tabular reinforcement learning methods, several significant concerns were raised.

**Resubmission Of Major Revision:**

The authors may consider submitting a major revision at a later time.